# Off-policy estimation with adaptively collected data: the power of online learning

**Jeonghwan Lee**
Department of Statistics
The University of Chicago
Chicago, IL 60637
jhlee97@uchicago.edu

**Cong Ma**
Department of Statistics
The University of Chicago
Chicago, IL 60637
congm@uchicago.edu

## Abstract

We consider estimation of a linear functional of the treatment effect from adaptively collected data. This problem finds a variety of applications including off-policy evaluation in contextual bandits, and estimation of the average treatment effect in causal inference. While a certain class of augmented inverse propensity weighting (AIPW) estimators enjoys desirable asymptotic properties including the semi-parametric efficiency, much less is known about their non-asymptotic theory with adaptively collected data. To fill in the gap, we first present generic upper bounds on the mean-squared error of the class of AIPW estimators that crucially depends on a sequentially weighted error between the treatment effect and its estimates. Motivated by this, we propose a general reduction scheme that allows one to produce a sequence of estimates for the treatment effect via online learning to minimize the sequentially weighted estimation error. To illustrate this, we provide three concrete instantiations in (1) the tabular case; (2) the case of linear function approximation; and (3) the case of general function approximation for the outcome model. We then provide a local minimax lower bound to show the instance-dependent optimality of the AIPW estimator using no-regret online learning algorithms.

## 1 Introduction

Estimating a linear functional of the treatment effect is of great importance in both causal inference and reinforcement learning (RL). For instance, in causal inference, one is interested in estimating the average treatment effect (ATE) [20] or their weighted variants, and in the literature of bandits and RL, one is interested in estimating the expected reward of a target policy [38, 64, 41, 37]. Two main challenges arise when tackling this problem:

- **Off-policy estimation**: Oftentimes, one needs to estimate the linear functional based on observational data collected from a behavior policy. This behavior policy may not match the desired distribution specified by the linear functional [42];

- **Adaptive data collection mechanism**: It is increasingly common for observational data to be adaptively collected due to the use of online algorithms (e.g., via contextual bandit algorithms [60, 33, 2, 52, 34]) in experimental design [67].

In this paper, we deal with two challenges simultaneously by investigating the estimation of a linear functional of the treatment effect from observational data that are collected adaptively. When the observational data is collected non-adaptively, i.e., in an i.i.d. manner, there is an extensive line of work [51, 49, 10, 24, 1, 27, 43, 6, 3, 64, 41] investigating the asymptotic and non-asymptotic theory of various estimators. Most notably are the study [6] that establishes the asymptotic efficiency of a family of semi-parametric estimators, and a more recent study [42] that undertakes a finite-sample

38th Conference on Neural Information Processing Systems (NeurIPS 2024).

analysis which uncovers the importance of a certain weighted $\ell_2$-norm when estimating the treatment effect. On the other hand, when it comes to adaptively collected data, most prior works [16, 67] focus on the asymptotic normality of the estimators, and do not discuss the finite-sample analysis of the estimators. In this paper, we aim to fill in this gap.

## 1.1 Main contributions

More specifically, we make the following three main contributions in this paper:

- First, we present generic finite-sample upper bounds on the mean-squared error of the class of *augmented inverse propensity weighting* (AIPW) estimators that crucially depends on a sequentially weighted error between the treatment effect and its estimates. This sequentially weighted estimation error demonstrates a clear effect of history-dependent behavior policies;

- Second, motivated by previous finding, we propose a general reduction scheme that allows one to form a sequence of estimates for the treatment effect via online learning to minimize the sequentially weighted estimation error. To demonstrate this, we provide three concrete instantiations in (1) the tabular case; (2) the case of linear function approximation; and (3) the case of general function approximation for the outcome model;

- In the end, we provide a local minimax lower bound to showcase the instance-dependent optimality of the AIPW estimator using no-regret online learning algorithms in the large-sample regime.

## 1.2 Related works

**Off-policy estimation with observational data**    Off-policy estimation in observational settings has been a central topic in statistics, operations research, causal inference, and RL. Here, we group a few prominent off-policy estimators into the following three categories: (i) *Model-based estimator*: often dubbed as the *direct method* (DM), whose key idea is to utilize observational data to learn a regression model that predicts outcomes for each state-action pair, and then average these model predictions [29, 10, 9, 39]. Due to model mis-specification, DM typically has a low variance but might lead to highly biased estimation results. (ii) *Inverse propensity weighting* (IPW): for the OPE task, IPW uses importance weighting to account for the distribution mismatch between the behavioral policy and the target policy [21, 55]. If the behavioral policy differs significantly from the target policy, then IPW can have an overly large variance (known as the *low overlap* issue) [23]. Typical remedies for this issue include propensity clipping [25, 57] or self-normalization [19, 58]. (iii) *Hybrid estimator*: some off-policy estimators (e.g., the doubly-robust (DR) estimator [10]) combine DM and IPW together to blend their complementary strengths [48, 10, 9, 59, 12, 56, 64]. A key asymptotic results in OPE is that the cross-fitted DR is $\sqrt{n}$-consistent and asymptotically efficient (that is, it attains the lowest possible asymptotic variance), even for the case where nuisance parameters are estimated at rates slower than $\sqrt{n}$-rates [6]. However, these methods still might be vulnerable to the low overlap issue especially for large or continuous action spaces. Thus, there has been a line of recent studies on OPE for large action spaces [13, 53, 44, 54] and OPE for continuous action space [28, 35, 63].

**Off-policy estimation with adaptively collected data**    A recent strand of works studied asymptotic theory of adaptive variants of the IPW and DR estimators (e.g., asymptotic normality, semi-parametric efficiency, and confidence intervals) [31, 8, 7] for adaptively collected data. However, in adaptive experiments, overlap between the behavioral policies and the target policy can deteriorate since the experimenter shifts the behavioral policies in response to what he/she observes (known as the *drifting overlap*) [67]. It may engender unacceptably large variances of the IPW and DR estimators. To address this large variance problem, there has been a recent strand of works investigating variance reduction strategies for the DR estimator based on shrinking importance weights toward one [4, 64, 57, 56], local stabilization [40, 69], and adaptive weighting [17, 67]. Recent studies on policy learning with adaptively collected data [68, 26] explored the adaptive weighting DR estimator for policy learning. In contrast with the majority of prior works on off-policy estimation with adaptively collected data that focus on asymptotic results, this paper aims at establishing non-asymptotic theory of the problem. While several researchers have been recently explored non-asymptotic results of the problem with an emphasis on uncertainty quantification [30, 65], we focus on analyses of estimation procedures of the off-policy value. As a majority of existing standard objects for uncertainty quantification, such as a confidence interval (CI), take a very static view of the world (e.g., it holds for a fixed sample size and

is not designed for interactive/adaptive data collection procedures), the aforementioned two papers [30, 65] instead study a more suitable statistical tool for such cases called a *confidence sequence*.

## 2  Problem formulation

We first formulate our problem using the language of contextual bandits: let $\mathbb{X}$, $\mathbb{A}$, and $\mathbb{Y} \subseteq \mathbb{R}$ denote the *context space*, the *action space*, and the *outcome space*, respectively. Denote by $\mathbb{O} := \mathbb{X} \times \mathbb{A} \times \mathbb{Y}$ the space of all possible context-action-outcome triples. In an adaptive experiment, one observes $n$ samples $\{(X_i, A_i, Y_i) \in \mathbb{O} : i \in [n]\}$ produced by the following data generating procedure [26, 68]: At each stage $i \in [n]$,

  (i) A context $X_i \in \mathbb{X}$ is independently sampled from a fixed *context distribution* $\Xi^*(\cdot) \in \Delta(\mathbb{X})$;

  (ii) There exists a *behavioral policy* $\Pi_i^*(\cdot, \cdot) : \mathbb{X} \times \mathbb{O}^{i-1} \to \Delta(\mathbb{A})$ that selects the $i$-th action as $A_i \mid X_i, \mathbf{O}_{i-1} \sim \Pi_i^* (\cdot \mid X_i, \mathbf{O}_{i-1})$, where $\mathbf{O}_i := (X_1, A_1, Y_1, \cdots, X_i, A_i, Y_i) \in \mathbb{O}^i$ for $i \in [n]$. As $\Pi_i^* (\cdot \mid X_i, \mathbf{O}_{i-1})$ may depend on previous observations, $\{(X_i, A_i, Y_i) : i \in [n]\}$ are no longer i.i.d.;

  (iii) Given a Markov kernel $\Gamma^*(\cdot, \cdot) : \mathbb{X} \times \mathbb{A} \to \Delta(\mathbb{Y})$, we assume that the outcome is generated according to $Y_i \sim \Gamma^* (\cdot \mid X_i, A_i)$. Moreover, the conditional mean of the outcome $Y_i \in \mathbb{Y}$ is specified as

$$\mathbb{E}\left[Y_i \mid X_i, A_i\right] = \int_{\mathbb{Y}} y \Gamma^* \left(\mathrm{d}y \mid X_i, A_i\right) = \mu^* \left(X_i, A_i\right),$$

  where the function $\mu^*(\cdot, \cdot) : \mathbb{X} \times \mathbb{A} \to \mathbb{R}$ is called the *treatment effect* (in causal inference) or the *reward function* (in bandit and RL literature). We note that the treatment effect $\mu^*$ is not revealed to the statistician. We also define the conditional variance function $\sigma^2(\cdot, \cdot) :$ $\mathbb{X} \times \mathbb{A} \to [0, +\infty]$ defined by $\sigma^2 (x, a) := \mathbb{E}\left[\left\{Y - \mu^* (X, A)\right\}^2 \Big| (X, A) = (x, a)\right]$, which is assumed to satisfy $\sigma^2(x, a) < +\infty$ for every state-action pair $(x, a) \in \mathbb{X} \times \mathbb{A}$.

At this moment, we assume the existence of $\sigma$-finite base measures $\lambda_{\mathbb{X}}(\cdot)$, $\lambda_{\mathbb{A}}(\cdot)$, and $\lambda_{\mathbb{Y}}(\cdot)$ over $\mathbb{X}$, $\mathbb{A}$, and $\mathbb{Y}$, resp., such that $\Xi^*(\cdot) \ll \lambda_{\mathbb{X}}(\cdot)$, $\Pi_i^* (\cdot \mid x, \mathbf{o}_{i-1}) \ll \lambda_{\mathbb{A}}(\cdot)$ for every $(x, \mathbf{o}_{i-1}) \in \mathbb{X} \times \mathbb{O}^{i-1}$ and $i \in [n]$, and $\Gamma^* (\cdot \mid x, a) \ll \lambda_{\mathbb{Y}}(\cdot)$ for all state-action pairs $(x, a) \in \mathbb{X} \times \mathbb{A}$. Here, the notation $\ll$ stands for the *absolute continuity* of measures. Our main goal is to estimate the *off-policy value* for any given target evaluation function $g(\cdot, \cdot) : \mathbb{X} \times \mathbb{A} \to \mathbb{R}$ defined as

$$\tau^* = \tau \left(\mathcal{I}^*\right) := \mathbb{E}_{X \sim \Xi^*} \left[\langle g(X, \cdot), \mu^*(X, \cdot)\rangle_{\lambda_{\mathbb{A}}}\right], \tag{1}$$

where $\mathcal{I}^* := (\Xi^*, \Gamma^*) \in \mathbb{I} := \Delta(\mathbb{X}) \times (\mathbb{X} \times \mathbb{A} \to \Delta(\mathbb{Y}))$ defines our *problem instance*. Throughout the paper, we assume that the propensity scores $\{\pi_i^* (X_i, \mathbf{O}_{i-1}; A_i) : i \in [n]\}$ are revealed, where $\pi_i^* (x, \mathbf{o}_{i-1}; \cdot) := \frac{\mathrm{d}\Pi_i^*(\cdot \mid x, \mathbf{o}_{i-1})}{\mathrm{d}\lambda_{\mathbb{A}}} : \mathbb{A} \to \mathbb{R}$.

As we mentioned earlier in Section 1, the estimation problem of a linear functional of the treatment effect $\mu^*$ turns out to be useful in both causal inference and RL in the following sense:

- **Estimation of average treatment effects**: We consider the binary action space $\mathbb{A} = \{0, 1\}$ equipped with the counting measure. The *average treatment effect* (ATE) in our problem setting is defined as the linear functional

$$\mathrm{ATE} := \mathbb{E}_{\mathcal{I}^*} \left[Y_i(1) - Y_i(0)\right] = \mathbb{E}_{X \sim \Xi^*} \left[\mu^* (X, 1) - \mu^* (X, 0)\right].$$

  Once we take the evaluation function as $g(x, a) = 2a - 1$, the ATE boils down to a particular case of the equation (1);

- **Off-policy evaluation (OPE) for contextual bandits**: Assume that a *target policy* $\Pi^{\mathrm{target}}(\cdot) :$ $\mathbb{X} \to \Delta(\mathbb{A})$ is given such that $\Pi^{\mathrm{target}} (\cdot \mid x) \ll \lambda_{\mathbb{A}}(\cdot)$ for every context $x \in \mathbb{X}$. For simplicity, let $\pi^{\mathrm{target}} (x, \cdot) := \frac{\mathrm{d}\Pi^{\mathrm{target}}(\cdot \mid x)}{\mathrm{d}\lambda_{\mathbb{A}}}$ denote the density function of the target policy for each context $x \in \mathbb{X}$. If we take $g(x, a) = \pi^{\mathrm{target}}(x, a)$, then the linear functional (1) corresponds to the value of the target policy $\Pi^{\mathrm{target}}$. This problem has been widely studied in the literature of bandits and RL, known as the *off-policy evaluation* (OPE).

We conclude this section by introducing notations that will be useful in later sections: let $\mathbb{P}_{\mathcal{I}}^i \in \Delta\left(\mathbb{O}^i\right)$ denote the law of the sample trajectory $\mathbf{O}_i$ under the sampling mechanism with a problem instance $\mathcal{I} = (\Xi, \Gamma) \in \mathbb{I}$. We denote the density function of $\mathbb{P}_{\mathcal{I}}^i \in \Delta\left(\mathbb{O}^i\right)$ with respect to the base measure $(\lambda_{\mathbb{X}} \otimes \lambda_{\mathbb{A}} \otimes \lambda_{\mathbb{Y}})^{\otimes i}$ by $p_{\mathcal{I}}^i(\cdot) : \mathbb{O}^i \to \mathbb{R}_+$. Lastly, we define the *k-th weighted $\ell_2$-norm* for $k \in [n]$ as

$$\|\varphi\|_{(k)}^2 := \frac{1}{k} \sum_{i=1}^{k} \mathbb{E}_{\mathcal{I}^*}\left[\frac{g^2\left(X_i, A_i\right) \varphi^2\left(X_i, A_i\right)}{\left(\pi_i^*\right)^2\left(X_i, \mathbf{O}_{i-1}; A_i\right)}\right] \tag{2}$$

for any function $\varphi(\cdot, \cdot) : \mathbb{X} \times \mathbb{A} \to \mathbb{R}$, together with the *k-th weighted $\ell_2$-space* by

$$\mathbb{L}_{(k)}^2 := \left\{\varphi(\cdot, \cdot) \in (\mathbb{X} \times \mathbb{A} \to \mathbb{R}) : \|\varphi\|_{(k)} < +\infty\right\}.$$

## 3 A class of AIPW estimators and non-asymptotic guarantees

The main objective of this section is to develop a meta-algorithm to tackle the estimation problem of the off-policy value (1), followed by some key rationale of the proposed procedure as a variance-reduction scheme of the standard *inverse propensity weighting* (IPW) estimator.

### 3.1 How can we reduce the variance of the IPW estimator?

Akin to [42], we consider a class of two-stage estimators obtained from simple perturbations of the IPW estimator. Given any collection $f := \left(f_i : \mathbb{X} \times \mathbb{O}^{i-1} \times \mathbb{A} \to \mathbb{R} : i \in [n]\right)$ of auxiliary functions, we consider the following *perturbed IPW estimator* $\hat{\tau}_n^f(\cdot) : \mathbb{O}^n \to \mathbb{R}$:

$$\hat{\tau}_n^f\left(\mathbf{o}_n\right) := \frac{1}{n} \sum_{i=1}^{n}\left\{\frac{g\left(x_i, a_i\right) y_i}{\pi_i^*\left(x_i, \mathbf{o}_{i-1}; a_i\right)} - f_i\left(x_i, \mathbf{o}_{i-1}, a_i\right) + \left\langle f_i\left(x_i, \mathbf{o}_{i-1}, \cdot\right), \pi_i^*\left(x_i, \mathbf{o}_{i-1}; \cdot\right)\right\rangle_{\lambda_{\mathbb{A}}}\right\}.$$

For each $i \in [n]$, let $\nu_i \in \Delta\left(\mathbb{X} \times \mathbb{O}^{i-1} \times \mathbb{A}\right)$ denote the joint distribution of $(X_i, \mathbf{O}_{i-1}, A_i)$ induced by the adaptive data collection procedure described in Section 2. Then, we arrive at the following result whose proof is deferred to Appendix B.1:

**Proposition 3.1.** *For any collection $f := \left(f_i \in L^2\left(\nu_i\right) : i \in [n]\right)$ of auxiliary deterministic functions, we have $\mathbb{E}_{\mathcal{I}^*}\left[\hat{\tau}_n^f\left(\mathbf{O}_n\right)\right] = \tau\left(\mathcal{I}^*\right)$. Furthermore, if*

$$\left\langle f_i\left(x, \mathbf{o}_{i-1}, \cdot\right), \pi_i^*\left(x, \mathbf{o}_{i-1}; \cdot\right)\right\rangle_{\lambda_{\mathbb{A}}} = 0, \; \forall\left(x, \mathbf{o}_{i-1}\right) \in \mathbb{X} \times \mathbb{O}^{i-1} \tag{3}$$

*for each $i \in [n]$, then*

$$n \cdot \mathrm{Var}_{\mathcal{I}^*}\left[\hat{\tau}_n^f\left(\mathbf{O}_n\right)\right] = \mathrm{Var}_{X \sim \Xi^*}\left[\left\langle g(X, \cdot), \mu^*(X, \cdot)\right\rangle_{\lambda_{\mathbb{A}}}\right] + \|\sigma\|_{(n)}^2 \tag{4}$$
$$+ \frac{1}{n} \sum_{i=1}^{n} \mathbb{E}_{\mathcal{I}^*}\left[\left\{\frac{g\left(X_i, A_i\right) \mu^*\left(X_i, A_i\right)}{\pi_i^*\left(X_i, \mathbf{O}_{i-1}; A_i\right)} - \left\langle g\left(X_i, \cdot\right), \mu^*\left(X_i, \cdot\right)\right\rangle_{\lambda_{\mathbb{A}}} - f_i\left(X_i, \mathbf{O}_{i-1}, A_i\right)\right\}^2\right].$$

From the decomposition (4) of the variance of the perturbed IPW estimate $\hat{\tau}_n^f\left(\mathbf{O}_n\right)$, one observes that the only term that depends on the collection of auxiliary functions $f$ is the third term. More importantly, the third term is equal to zero if and only if

$$f_i\left(x, \mathbf{o}_{i-1}, a\right) = f_i^*\left(x, \mathbf{o}_{i-1}, a\right) := \frac{g\left(x, a\right) \mu^*\left(x, a\right)}{\pi_i^*\left(x, \mathbf{o}_{i-1}; a\right)} - \left\langle g(x, \cdot), \mu^*(x, \cdot)\right\rangle_{\lambda_{\mathbb{A}}}. \tag{5}$$

The collection of minimizing functions $f^* := \left(f_i^* \in L^2\left(\nu_i\right) : i \in [n]\right)$ yields the *oracle estimator* $\hat{\tau}_n^{f^*}(\cdot) : \mathbb{O}^n \to \mathbb{R}$

$$\hat{\tau}_n^{f^*}\left(\mathbf{O}_n\right) = \frac{1}{n} \sum_{i=1}^{n}\left\{\frac{g\left(X_i, A_i\right)\left\{Y_i - \mu^*\left(X_i, A_i\right)\right\}}{\pi_i^*\left(X_i, \mathbf{O}_{i-1}; A_i\right)} + \left\langle g\left(X_i, \cdot\right), \mu^*\left(X_i, \cdot\right)\right\rangle_{\lambda_{\mathbb{A}}}\right\}, \tag{6}$$

whose variance is given by

$$n \cdot \mathrm{Var}_{\mathcal{I}^*}\left[\hat{\tau}_n^{f^*}\left(\mathbf{O}_n\right)\right] = v_*^2 := \mathrm{Var}_{X \sim \Xi^*}\left[\left\langle g(X, \cdot), \mu^*(X, \cdot)\right\rangle_{\lambda_{\mathbb{A}}}\right] + \|\sigma\|_{(n)}^2. \tag{7}$$

---

**Algorithm 1** Meta-algorithm: augmented inverse propensity weighting (AIPW) estimator.

---

**Require:** the dataset $\mathcal{D} = \{(X_i, A_i, Y_i) \in \mathbb{O} : i \in [n]\}$ and an evaluation function $g : \mathbb{X} \times \mathbb{A} \to \mathbb{R}$.

1: For each step $i \in [n]$, we compute an estimate $\hat{\mu}_i(\mathbf{O}_{i-1}) \in (\mathbb{X} \times \mathbb{A} \to \mathbb{R})$ of the treatment effect based on the sample trajectory $\mathbf{O}_{i-1}$ up to the $(i-1)$-th step. `// Implement Algorithm 2 as a subroutine;`

2: Consider the AIPW estimator (a.k.a., the *doubly-robust* (DR) estimator) $\hat{\tau}_n^{\mathsf{AIPW}}(\cdot) : \mathbb{O}^n \to \mathbb{R}$:

$$\hat{\tau}_n^{\mathsf{AIPW}}(\mathbf{o}_n) := \frac{1}{n} \sum_{i=1}^{n} \hat{\Gamma}_i(\mathbf{o}_i), \tag{8}$$

where the objects being averaged are the AIPW scores $\hat{\Gamma}_i(\cdot) : \mathbb{O}^i \to \mathbb{R}$ is defined by

$$\hat{\Gamma}_i(\mathbf{o}_i) := \frac{g(x_i, a_i)}{\pi_i^*(x_i, \mathbf{o}_{i-1}; a_i)} \{y_i - \hat{\mu}_i(\mathbf{o}_{i-1})(x_i, a_i)\} + \langle g(x_i, \cdot), \hat{\mu}_i(\mathbf{o}_{i-1})(x_i, \cdot)\rangle_{\lambda_{\mathbb{A}}}. \tag{9}$$

3: **return** the AIPW estimate $\hat{\tau}_n^{\mathsf{AIPW}}(\mathbf{O}_n)$.

---

### 3.2 The class of augmented IPW estimators

Since the treatment effect $\mu^*$ is not revealed to the statistician in (6), it is impossible to exactly compute the oracle estimate $\hat{\tau}_n^{f^*}(\cdot) : \mathbb{O}^n \to \mathbb{R}$ using only the observational dataset $\mathbf{O}_n$. Therefore, a natural remedy would be the following two-stage procedure, which is referred to as the *augmented inverse propensity weighting* (AIPW) estimator or the *doubly-robust* (DR) estimator [10, 50, 61, 17, 67, 22]: (i) we first compute a sequence of estimates $\{\hat{\mu}_i(\mathbf{O}_{i-1}) \in (\mathbb{X} \times \mathbb{A} \to \mathbb{R}) : i \in [n]\}$ of the treatment effect $\mu^*$; and then (ii) we plug-in these estimates to the equation (6) to construct an approximation to the ideal estimate $\hat{\tau}_n^{f^*}(\mathbf{O}_n)$. We summarize this two-stage procedure in Algorithm 1.

We pause here to compare our problem setting and algorithms with the most relevant work [42]. We focus on off-policy estimation with adaptively collected data, which is technically more challenging compared to i.i.d. data considered in [42]. In the case with i.i.d. data, [42] proposed a natural approach to construct a class of two-stage estimators as follows: (a) compute an estimate $\hat{\mu}$ of the treatment effect $\mu^*$ utilizing part of the dataset; and (b) substitute this estimate in the equation (6) of the oracle estimator. Note that the authors use the *cross-fitting approach* [5, 6], which allows to make full use of data to maintain efficiency and statistical power of machine learning algorithms for estimation of nuisance parameters while reducing overfitting bias. However, the cross-fitting strategy heavily relies on the i.i.d. nature of the data collection mechanism and therefore one cannot use it in the setting with adaptively collected data. Instead, we construct an estimate $\hat{\mu}_i$ of the treatment effect $\mu^*$ based on the sample trajectory $\mathbf{O}_{i-1}$ at each stage and then substitute these estimates in the equation (6). This is one of main contributions to address the adaptive nature of our data generating mechanism. We will make use of the framework of online learning to construct a sequence of estimates for the treatment effect $\mu^*$.

### 3.3 Theoretical guarantees of Algorithm 1

In this section, we provide statistical guarantees for the class of AIPW estimators for dealing with the estimation problem of the off-policy value (1). The main result of this section can be summarized as the following non-asymptotic upper bound on the mean-squared error (MSE) of Algorithm 1:

**Theorem 3.1** (Non-asymptotic upper bound on the MSE of the AIPW estimator). *For any sequence of estimates $\{\hat{\mu}_i(\mathbf{O}_{i-1}) \in (\mathbb{X} \times \mathbb{A} \to \mathbb{R}) : i \in [n]\}$ for the treatment effect $\mu^*$, the AIPW estimator (8) has the MSE bounded above by*

$$\mathbb{E}_{\mathcal{I}^*}\left[\left\{\hat{\tau}_n^{\mathsf{AIPW}}(\mathbf{O}_n) - \tau(\mathcal{I}^*)\right\}^2\right]$$
$$\leq \frac{1}{n}\left\{v_*^2 + \frac{1}{n}\sum_{i=1}^{n}\mathbb{E}\left[\frac{g^2(X_i, A_i)\{\hat{\mu}_i(\mathbf{O}_{i-1})(X_i, A_i) - \mu^*(X_i, A_i)\}^2}{(\pi_i^*)^2(X_i, \mathbf{O}_{i-1}; A_i)}\right]\right\}. \tag{10}$$

Note that the non-asymptotic upper bound (10) on the MSE for the class of AIPW estimators (8) consists of two terms, both of which have natural interpretations. The first term $v_*^2$ corresponds to the

---

**Algorithm 2** Online non-parametric regression protocol for estimation of the treatment effect.

**Require:** the number of rounds $n \in \mathbb{N}$.
 1: **for** $i = 1, 2, \cdots, n$, **do**
 2:     The learner selects a point $\hat{\mu}_i (\mathbf{O}_{i-1}) \in (\mathbb{X} \times \mathbb{A} \to \mathbb{R})$ based on the sample trajectory $\mathbf{O}_{i-1}$;
 3:     The environment then picks a loss function $l_i(\cdot) : (\mathbb{X} \times \mathbb{A} \to \mathbb{R}) \to \mathbb{R}$ defined as

$$l_i(\mu) := \frac{g^2 (X_i, A_i)}{(\pi_i^*)^2 (X_i, \mathbf{O}_{i-1}; A_i)} \{Y_i - \mu (X_i, A_i)\}^2, \ \forall \mu(\cdot, \cdot) \in (\mathbb{X} \times \mathbb{A} \to \mathbb{R}). \quad (14)$$

 4: **end for**
 5: **return** the sequence of estimates $\{\hat{\mu}_i (\mathbf{O}_{i-1}) \in (\mathbb{X} \times \mathbb{A} \to \mathbb{R}) : i \in [n]\}$ of the treatment effect.

---

optimal variance (7) achievable by the oracle estimator, and the second term

$$\frac{1}{n} \sum_{i=1}^n \mathbb{E}_{\mathcal{I}^*} \left[ \frac{g^2 (X_i, A_i) \{\hat{\mu}_i (\mathbf{O}_{i-1}) (X_i, A_i) - \mu^* (X_i, A_i)\}^2}{(\pi_i^*)^2 (X_i, \mathbf{O}_{i-1}; A_i)} \right] \quad (11)$$

measures the average estimation error of the estimates $\{\hat{\mu}_i (\mathbf{O}_{i-1}) \in (\mathbb{X} \times \mathbb{A} \to \mathbb{R}) : i \in [n]\}$ of $\mu^*$. Of primary interest to us is a subsequent upper bounding argument based on the MSE bound (10) in the finite sample regime: in particular, to minimize the RHS of (10), one needs to choose a sequence of estimates $\{\hat{\mu}_i (\mathbf{O}_{i-1}) \in (\mathbb{X} \times \mathbb{A} \to \mathbb{R}) : i \in [n]\}$ which minimizes the second term (11).

### 3.4 Reduction to online non-parametric regression

Let us now focus on constructing a sequence of estimates $\{\hat{\mu}_i (\mathbf{O}_{i-1}) \in (\mathbb{X} \times \mathbb{A} \to \mathbb{R}) : i \in [n]\}$ of the treatment effect and upper bounding the estimation error (11) in the MSE bound (10). To this end, we borrow ideas from the literature of online non-parametric regression [45].

To begin with, we consider an $n$-round turn-based game between the learner and the environment; see Algorithm 2 for the details. Then, one can readily observe for any $\mu(\cdot, \cdot) : \mathbb{X} \times \mathbb{A} \to \mathbb{R}$, we have

$$\begin{aligned}
& \mathbb{E}_{\mathcal{I}^*} \left[ l_i(\mu) | (\mathcal{H}_{i-1}, X_i, A_i) \right] \\
&= \frac{g^2 (X_i, A_i)}{(\pi_i^*)^2 (X_i, \mathbf{O}_{i-1}; A_i)} \left[ \sigma^2 (X_i, A_i) + \{\mu (X_i, A_i) - \mu^* (X_i, A_i)\}^2 \right].
\end{aligned} \quad (12)$$

In the current turn-based game, our natural goal is to minimize the learner's static regret against the *best fixed action in hindsight* belonging to a pre-specified function class $\mathcal{F} \subseteq (\mathbb{X} \times \mathbb{A} \to \mathbb{R})$:

$$\text{Regret} (n, \mathcal{F}; \mathcal{A}) := \sum_{i=1}^n l_i \{\hat{\mu}_i (\mathbf{O}_{i-1})\} - \inf_{\mu \in \mathcal{F}} \sum_{i=1}^n l_i(\mu), \quad (13)$$

where $\mathcal{A}$ denotes the learner's online non-parametric regression algorithm that returns a sequence of estimates $\{\hat{\mu}_i (\mathbf{O}_{i-1}) : i \in [n]\}$ for the treatment effect. Then, one can establish the following oracle inequality that demystifies a relationship between estimation problem of the off-policy value and the online non-parametric regression protocol. See Appendix B.3 for the proof.

**Theorem 3.2** (Oracle inequality for the class of AIPW estimators)**.** *The AIPW estimator* (8) *using the sequence of estimates* $\{\hat{\mu}_i (\mathbf{O}_{i-1}) \in (\mathbb{X} \times \mathbb{A} \to \mathbb{R}) : i \in [n]\}$ *of the treatment effect* $\mu^*$ *produced by the online non-parametric regression algorithm* $\mathcal{A}$ *enjoys the following upper bound on the MSE:*

$$\begin{aligned}
& \mathbb{E}_{\mathcal{I}^*} \left[ \{\hat{\tau}_n^{\mathsf{AIPW}} (\mathbf{O}_n) - \tau (\mathcal{I}^*)\}^2 \right] \\
& \leq \frac{1}{n} \left( v_*^2 + \frac{1}{n} \mathbb{E}_{\mathcal{I}^*} \left[ \text{Regret} (n, \mathcal{F}; \mathcal{A}) \right] + \inf \left\{ \|\mu - \mu^*\|_{(n)}^2 : \mu \in \mathcal{F} \right\} \right).
\end{aligned} \quad (15)$$

A few remarks are in order. Apart from the optimal variance $v_*^2$, the RHS of the bound (15) contains two additional terms: (i) the expected regret relative to the number of rounds $n$, where the expected value is taken over $\mathbf{O}_n \sim \mathbb{P}_{\mathcal{I}^*}^n (\cdot)$; and (ii) the approximation error under the $\|\cdot\|_{(n)}$-norm. Given any fixed function class $\mathcal{F} \subseteq (\mathbb{X} \times \mathbb{A} \to \mathbb{R})$, if we consider the large sample size regime, i.e., the sample

size $n$ is sufficiently large, then one can see that the asymptotic variance of the AIPW estimator (8) is asymptotically the same as $v_*^2 + \inf\left\{\|\mu - \mu^*\|_{(n)}^2 : \mu \in \mathcal{F}\right\}$, provided that the online non-parametric regression algorithm $\mathcal{A}$ exhibits a *no-regret learning dynamics*, i.e., $\mathbb{E}_{\mathcal{I}^*}\left[\text{Regret}(n, \mathcal{F}; \mathcal{A})\right] = o(n)$ as $n \to \infty$. Consequently, the AIPW estimator (8) may suffer from an efficiency loss which depends on how well the unknown treatment effect $\mu^*$ can be approximated by a member of the function class $\mathcal{F} \subseteq (\mathbb{X} \times \mathbb{A} \to \mathbb{R})$ under the $\|\cdot\|_{(n)}$-norm. Hence, any contribution to the MSE bound of the AIPW estimator (8) *in addition to* the efficient variance $v_*^2$ primarily relies on the approximation error associated with approximating the treatment effect $\mu^*$ utilizing a provided function class $\mathcal{F}$.

### 3.5 Consequences for particular outcome models

The main goal of this section is to illustrate the consequences of our general theory developed in Section 3 so far for several concrete classes of outcome models. Throughout this section, we consider the case for which $\mathbb{Y} = [-L, L]$ for some constant $L \in (0, +\infty)$, and impose the following condition:

**Assumption 1** (Strict overlap condition). The likelihood ratios are uniformly bounded by a universal constant $B \in (0, +\infty)$, i.e., for every $i \in [n]$,

$$\left|\frac{g(X_i, A_i)}{\pi_i^*(X_i, \mathbf{O}_{i-1}; A_i)}\right| \leq B \quad \mathbb{P}_{\mathcal{I}^*}^n\text{-almost surely.} \tag{16}$$

We note that Assumption 1 is often referred to as the *strict overlap condition* in the literature of causal inference [20, 32, 66, 36, 11]. At this point, we emphasize that Assumption 1 is necessary to produce main consequences of the oracle inequality for the class of AIPW estimators (Theorem 3.2) that we discuss in the ensuing subsections: Theorems 3.3, 3.4, and the arguments throughout Appendix B.6.

#### 3.5.1 Tabular case of the outcome model

We embark on our discussion about the consequences of our theory established in Sections 3.3 and 3.4 for one of the simplest case of the outcome model satisfying the following assumption.

**Assumption 2** (Tabular setting of the outcome model). The state-action space $\mathbb{X} \times \mathbb{A}$ is a finite set.

If we compute the gradient of the loss function (14), we have

$$\nabla l_i(\mu) = \frac{2g^2(X_i, A_i)}{(\pi^*)^2(X_i, \mathbf{O}_{i-1}; A_i)}\left\{\mu(X_i, A_i) - Y_i\right\}\delta_{(X_i, A_i)}, \quad \forall \mu \in \mathbb{R}^{\mathbb{X} \times \mathbb{A}}, \tag{17}$$

where $\delta_{(X_i, A_i)} \in \mathbb{R}^{\mathbb{X} \times \mathbb{A}}$ is the point-mass vector at the $i$-th state-action pair in the sample trajectory, i.e., $\delta_{(X_i, A_i)}(x, a) := 1$ if $(x, a) = (X_i, A_i)$; $\delta_{(X_i, A_i)}(x, a) := 0$ otherwise.

---

**Algorithm 3** Online gradient descent (OGD) algorithm for the finite state-action space.

---

**Require:** the function class $\mathcal{F} \subseteq [-L, L]^{\mathbb{X} \times \mathbb{A}}$, the total number of rounds $n \in \mathbb{N}$, and a sequence of learning rates $\{\eta_i \in (0, +\infty) : i \in [n-1]\}$.
1: We first choose an initial point $\hat{\mu}_1(\varnothing) \in \mathcal{F}$ arbitrarily;
2: **for** $i = 1, 2, \cdots, n-1$, **do**
3:     Observe a triple $(X_i, A_i, Y_i) \in \mathbb{O}$;
4:     Update $\hat{\mu}_{i+1}(\mathbf{O}_i) \in \mathcal{F}$ according to the following OGD update rule:

$$\hat{\mu}_{i+1}(\mathbf{O}_i) = \Pi_{\mathcal{F}}\left[\hat{\mu}_i(\mathbf{O}_{i-1}) - \eta_i \nabla l_i\{\hat{\mu}_i(\mathbf{O}_{i-1})\}\right]$$
$$= \Pi_{\mathcal{F}}\left[\hat{\mu}_i(\mathbf{O}_{i-1}) - \frac{2\eta_i \cdot g^2(X_i, A_i)}{(\pi_i^*)^2(X_i, \mathbf{O}_{i-1}; A_i)}\left\{\hat{\mu}_i(\mathbf{O}_{i-1}) - Y_i\right\}\delta_{(X_i, A_i)}\right], \tag{18}$$

    where $\Pi_{\mathcal{F}}[\cdot] : \mathbb{R}^{\mathbb{X} \times \mathbb{A}} \to \mathcal{F}$ denotes the projection map of $\mathbb{R}^{\mathbb{X} \times \mathbb{A}}$ onto the function space $\mathcal{F}$.
5: **end for**
6: **return** the sequence of estimates $\{\hat{\mu}_i(\mathbf{O}_{i-1}) \in \mathcal{F} : i \in [n]\}$ of the treatment effect $\mu^*$.

---

Now, it is time to put forward an online contextual learning algorithm aimed at producing a sequence of estimates of $\mu^*$ with a no-regret learning guarantee. For the tabular case, the online non-parametric

regression problem can be resolved through standard online convex optimization (OCO) algorithms. In particular, we employ the online gradient descent (OGD) algorithm (see Algorithm 3) as a sub-routine of Algorithm 1. By leveraging standard results on regret analysis of OCO algorithms, one can obtain the following regret bound, which guarantees a no-regret learning dynamics of Algorithm 3.

**Theorem 3.3** (Regret guarantee of Algorithm 3). *Under Assumptions 1 and 2, the OGD algorithm (Algorithm 3) with learning rates $\left\{ \eta_i := \frac{\mathrm{diam}(\mathcal{F})}{4LB^2\sqrt{i}} : i \in [n] \right\}$ guarantees*

$$\mathrm{Regret}\,(n, \mathcal{F}; \mathrm{OGD}) \leq 6LB^2 \mathrm{diam}(\mathcal{F}) \cdot \sqrt{n} \quad \mathbb{P}^n_{\mathcal{I}^*}\text{-almost surely}, \tag{19}$$

*where* $\mathrm{diam}(\mathcal{F}) := \sup\left\{ \|\mu\|_2 : \mu \in \mathcal{F} \right\}$ *denotes the diameter of* $\mathcal{F} \subseteq [-L, L]^{\mathbb{X} \times \mathbb{A}}$.

See Appendix B.4 for the proof of Theorem 3.3. Combining the regret guarantee (19) of Algorithm 3 together with the MSE bound (15) in Theorem 3.2, one can establish a concrete upper bound on the MSE of the AIPW estimator (8) by utilizing Algorithm 3 to produce a sequence of estimates for the treatment effect $\mu^*$.

### 3.5.2 Linear function approximation

We next move on to outcome models where the state-action space $\mathbb{X} \times \mathbb{A}$ can be infinite. We begin with the simplest case: the class of linear outcome functions. Let $\phi(\cdot, \cdot) : \mathbb{X} \times \mathbb{A} \to \mathbb{R}^d$ be a *known feature map* such that $\sup\left\{ \|\phi(x, a)\|_2 : (x, a) \in \mathbb{X} \times \mathbb{A} \right\} \leq 1$, and we consider the functions that are linear in this representation: $f_{\boldsymbol{\theta}}(\cdot, \cdot) : \mathbb{X} \times \mathbb{A} \to \mathbb{R}$, where $f_{\boldsymbol{\theta}}(x, a) := \boldsymbol{\theta}^\top \phi(x, a)$ for some parameter vector $\boldsymbol{\theta} \in \mathbb{R}^d$. Given a radius $R > 0$, we define the function class

$$\mathcal{F}_{\mathrm{lin}} := \left\{ f_{\boldsymbol{\theta}}(\cdot, \cdot) \in (\mathbb{X} \times \mathbb{A} \to \mathbb{R}) : \boldsymbol{\theta} \in \Theta := \overline{\mathbb{B}\,(\mathbf{0}_d; R)} \right\}, \tag{20}$$

where $\overline{\mathbb{B}\,(\mathbf{0}_d; R)} := \left\{ \mathbf{u} \in \mathbb{R}^d : \|\mathbf{u}\|_2 \leq R \right\}$. With this linear function approximation framework, let us consider the following OCO model: at the $i$-th stage,

(i) the learner first chooses a point $\hat{\theta}_i\,(\mathbf{O}_{i-1}) \in \Theta$;

(ii) the environment then picks a loss function $\mathcal{L}_i(\cdot) : \Theta \to \mathbb{R}$ defined as

$$\mathcal{L}_i(\boldsymbol{\theta}) := \frac{g^2\,(X_i, A_i)}{(\pi_i^*)^2\,(X_i, \mathbf{O}_{i-1}; A_i)} \left\{ Y_i - \boldsymbol{\theta}^\top \phi\,(X_i, A_i) \right\}^2, \ \forall \boldsymbol{\theta} \in \Theta, \tag{21}$$

and our goal is to produce a sequence of estimates $\left\{ \hat{\mu}_i\,(\mathbf{O}_{i-1}) := \left\{ \hat{\boldsymbol{\theta}}_i\,(\mathbf{O}_{i-1}) \right\}^\top \phi \in \mathcal{F}_{\mathrm{lin}} : i \in [n] \right\}$ for the treatment effect $\mu^*$ after $n$ rounds of the above-mentioned OCO model which minimizes the learner's regret against the *best fixed action in hindsight*:

$$\begin{aligned}
\mathrm{Regret}\,(n, \mathcal{F}_{\mathrm{lin}}; \mathcal{A}) &= \sum_{i=1}^n l_i\left\{ \hat{\mu}_i\,(\mathbf{O}_{i-1}) \right\} - \inf\left\{ \sum_{i=1}^n l_i(\mu) : \mu \in \mathcal{F} \right\} \\
&= \sum_{i=1}^n \mathcal{L}_i\left\{ \hat{\boldsymbol{\theta}}_i\,(\mathbf{O}_{i-1}) \right\} - \inf\left\{ \sum_{i=1}^n \mathcal{L}_i(\boldsymbol{\theta}) : \boldsymbol{\theta} \in \Theta \right\},
\end{aligned}$$

where $\mathcal{A}$ is the learner's OCO algorithm whose output is a sequence $\left\{ \hat{\boldsymbol{\theta}}_i\,(\mathbf{O}_{i-1}) \in \Theta : i \in [n] \right\}$ of parameters. If we compute the gradient of the loss function (21), one has

$$\nabla_{\boldsymbol{\theta}} \mathcal{L}_i(\boldsymbol{\theta}) = \frac{2g^2\,(X_i, A_i)}{(\pi_i^*)^2\,(X_i, \mathbf{O}_{i-1}; A_i)} \left\{ \boldsymbol{\theta}^\top \phi\,(X_i, A_i) - Y_i \right\} \phi\,(X_i, A_i). \tag{22}$$

For the current linear function approximation setting, we implement the OGD algorithm (Algorithm 4) as a sub-routine of Algorithm 1. By using the same arguments as in Section 3.5.1, one can reproduce the following regret guarantee of Algorithm 4 whose proof is available at Appendix B.5.

**Theorem 3.4** (Regret guarantee of Algorithm 4). *With Assumption 1, the OGD algorithm (Algorithm 4) with learning rates $\left\{ \eta_i := \frac{R}{B^2(L+R)\sqrt{i}} : i \in [n] \right\}$ guarantees*

$$\mathrm{Regret}\,(n, \mathcal{F}_{\mathrm{lin}}; \mathrm{OGD}) \leq 6B^2 R(L+R)\sqrt{n} \quad \mathbb{P}^n_{\mathcal{I}^*}\text{-almost surely}. \tag{24}$$

---

**Algorithm 4** Online gradient descent (OGD) algorithm for linear function approximation.

---

**Require:** the radius $R > 0$ of the parameter space, the number of rounds $n \in \mathbb{N}$, and a sequence of learning rates $\{\eta_i \in (0, +\infty) : i \in [n-1]\}$.

1: We first choose an arbitrary initial point $\hat{\boldsymbol{\theta}}_1(\varnothing) \in \Theta$, where $\Theta := \overline{\mathbb{B}(\mathbf{0}_d; R)}$;
2: **for** $i = 1, 2, \cdots, n-1$, **do**
3:      Observe a triple $(X_i, A_i, Y_i) \in \mathbb{O}$;
4:      Update $\hat{\boldsymbol{\theta}}_{i+1}(\mathbf{O}_i) \in \Theta$ according to the following OGD update rule:

$$\hat{\boldsymbol{\theta}}_{i+1}(\mathbf{O}_i) = \Pi_\Theta \left[ \hat{\boldsymbol{\theta}}_i(\mathbf{O}_{i-1}) - \eta_i \nabla_{\boldsymbol{\theta}} \mathcal{L}_i \left\{ \hat{\boldsymbol{\theta}}_i(\mathbf{O}_{i-1}) \right\} \right], \tag{23}$$

     where $\Pi_\Theta[\cdot] : \mathbb{R}^d \to \Theta$ denotes the projection map of $\mathbb{R}^d$ onto the parameter space $\Theta$.
5: **end for**
6: **return** the estimates $\left\{ \hat{\mu}_i(\mathbf{O}_{i-1}) := \left\{ \hat{\boldsymbol{\theta}}_i(\mathbf{O}_{i-1}) \right\}^\top \phi \in \mathcal{F}_{\text{lin}} : i \in [n] \right\}$ of the treatment effect.

---

**General function approximation**    Lastly, we demonstrate the consequences of our general theory established in Sections 3.3 and 3.4 for the case of general function approximation: the function class $\mathcal{F} \subseteq (\mathbb{X} \times \mathbb{A} \to [-L, L])$ can be arbitrarily chosen. Our further discussion this case heavily relies on the basic theory of online non-parametric regression from [45] whose technical details are rather long and complicated. So, we defer our detailed inspection on the case of general function approximation to Appendix B.6.

## 4   Lower bounds: local minimax risk

We turn our attention to a local minimax lower bound for estimating the off-policy value $\tau^* = \tau(\mathcal{I}^*)$. Here, we aim at establishing lower bounds that hold uniformly over all estimators that are permitted to know both the propensity scores $\{\pi_i^*(X_i, \mathbf{O}_{i-1}; A_i) : i \in [n]\}$ and the evaluation function $g$. We assume the existence of a constant $K \geq 1$ and *reference Markov policies* $\{\overline{\Pi}_i : \mathbb{X} \to \Delta(\mathbb{A}) : i \in [n]\}$ such that $\overline{\Pi}_i(\cdot \,|\, x) \ll \lambda_\mathbb{A}(\cdot)$ for $(x, i) \in \mathbb{X} \times [n]$, and

$$\frac{1}{K} \leq \frac{\overline{\pi}_i(x, a)}{\pi_i^*(x, \mathbf{o}_{i-1}; a)} \leq K \tag{25}$$

for all $(x, \mathbf{o}_{i-1}, a) \in \mathbb{X} \times \mathbb{O}^{i-1} \times \mathbb{A}$, where $\overline{\pi}_i(x, \cdot) := \frac{\mathrm{d}\overline{\Pi}_i(\cdot \,|\, x)}{\mathrm{d}\lambda_\mathbb{A}} : \mathbb{A} \to \mathbb{R}_+$ for each context $x \in \mathbb{X}$. Proximity of behavioral policies to certain Markov policies is often assumed under adaptive data collection procedures. For instance, in *Theorem 1* of [67], the authors assumed that the sequence of behavior policies is *eventually Markov*; see the equation (8) therein.

### 4.1   Instance-dependent local minimax lower bounds

Given any problem instance $\mathcal{I}^* = (\Xi^*, \Gamma^*) \in \mathbb{I}$ and an error function $\delta : \mathbb{X} \times \mathbb{A} \to \mathbb{R}_+$, we consider the following local neighborhoods:

$$\mathcal{N}(\Xi^*) := \left\{ \Xi \in \Delta(\mathbb{X}) : \mathrm{KL}(\Xi \,\|\, \Xi^*) \leq \frac{1}{n} \right\};$$

$$\mathcal{N}_\delta(\Gamma^*) := \left\{ \Gamma \in (\mathbb{X} \times \mathbb{A} \to \Delta(\mathbb{Y})) : |\mu(\Gamma)(x, a) - \mu(\Gamma^*)(x, a)| \leq \delta(x, a), \ \forall (x, a) \in \mathbb{X} \times \mathbb{A} \right\},$$

where for any given $\Gamma : \mathbb{X} \times \mathbb{A} \to \Delta(\mathbb{Y})$, let $\mu(\Gamma)(x, a) := \int_\mathbb{Y} y\Gamma(\mathrm{d}y \,|\, x, a)$ for each $(x, a) \in \mathbb{X} \times \mathbb{A}$. Our goal is to lower bound the following *local minimax risk*:

$$\mathcal{M}_n(\mathcal{C}_\delta(\mathcal{I}^*)) := \inf_{\hat{\tau}_n(\cdot) : \mathbb{O}^n \to \mathbb{R}} \left( \sup_{\mathcal{I} \in \mathcal{C}_\delta(\mathcal{I}^*)} \mathbb{E}_\mathcal{I} \left[ \{\hat{\tau}_n(\mathbf{O}_n) - \tau(\mathcal{I})\}^2 \right] \right), \tag{26}$$

where $\mathcal{C}_\delta(\mathcal{I}^*) := \mathcal{N}(\Xi^*) \times \mathcal{N}_\delta(\Gamma^*) \subseteq \mathbb{I}$. We now specify some assumptions necessary for lower bounding the local minimax risk (26). Prior to this, we introduce a new important notation: given any random variable $Y \in \mathbb{L}^4(\Omega, \mathcal{F}, \mathbb{P})$ defined on a probability space $(\Omega, \mathcal{F}, \mathbb{P})$, its $(2, 4)$-*moment ratio* is defined as $\|Y\|_{2 \to 4} := \frac{\sqrt{\mathbb{E}[Y^4]}}{\mathbb{E}[Y^2]}$.

**Assumption 3.** Let $h(x) := \langle g(x, \cdot), \mu^*(x, \cdot) \rangle_{\lambda_{\mathbb{A}}} - \mathbb{E}_{X \sim \Xi^*} \left[ \langle g(X, \cdot), \mu^*(X, \cdot) \rangle_{\lambda_{\mathbb{A}}} \right]$. We assume that $H_{2 \to 4} := \|h\|_{2 \to 4} = \frac{\sqrt{\mathbb{E}_{X \sim \Xi^*}[h^4(X)]}}{\mathbb{E}_{X \sim \Xi^*}[h^2(X)]} < +\infty$.

We next make an assumption on a lower bound on the *local neighborhood size*:

**Assumption 4.** The neighborhood function $\delta(\cdot, \cdot) : \mathbb{X} \times \mathbb{A} \to \mathbb{R}_+$ satisfies the lower bound

$$\sqrt{n} \cdot \delta(x, a) \geq \frac{|g(x, a)| \, \sigma^2(x, a)}{\overline{\pi}_i(x, a) \, \|\sigma\|_{(n)}} \tag{27}$$

for all $(x, a, i) \in \mathbb{X} \times \mathbb{A} \times [n]$.

We note that Assumptions 3 and 4 are analogues of Assumptions (MR) and (LN) considered in [42], respectively, for the case of adaptively collected data. Under these assumptions, one can prove the following lower bound on the local minimax risk over $\mathcal{C}_\delta(\mathcal{I}^*)$:

**Theorem 4.1.** *Under Assumptions 3 and 4, the local minimax risk over $\mathcal{C}_\delta(\mathcal{I}^*)$ is lower bounded by*

$$\mathcal{M}_n(\mathcal{C}_\delta(\mathcal{I}^*)) \geq \mathcal{C}(K) \cdot \frac{v_*^2}{n}, \tag{28}$$

*where $\mathcal{C}(K) > 0$ is a universal constant that only depends on the data coverage constant $K \geq 1$ of the reference Markov policies $\left\{ \overline{\Pi}_i(\cdot) : \mathbb{X} \to \Delta(\mathbb{A}) : i \in [n] \right\}$ defined in* (25).

The proof of Theorem 4.1 can be found in Appendix C.1. This result delivers a key message: the term $\frac{v_*^2}{n}$ including the sequentially weighted $\ell_2$-norm is indeed the fundamental limit for estimating the linear functional based on adaptively collected data. Our results can be viewed as a generalization of those developed in [42] for the case of i.i.d. data.

## Acknowledgments and Disclosure of Funding

Jeonghwan Lee was partially supported by the Kwanjeong Educational Foundation. Cong Ma was partially supported by the National Science Foundation via grant DMS-2311127.

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

# A   Some elementary inequalities and their proofs

The following lemma is useful for the truncation arguments used in the proofs of our local minimax lower bounds. In particular, it enables to make small modifications on a pair of probability measures by conditioning on *good events* of each probability measure, without inducing an overly large change in the total variation distance.

**Lemma A.1.** *Let $(\mu, \nu)$ be a pair of probability measures defined on a common sample space $(\Omega, \mathcal{F})$, and consider any two events $A, B \in \mathcal{F}$ satisfying $\min \{\mu(A), \nu(B)\} \geq 1 - \epsilon$ for some $\epsilon \in \left[0, \frac{1}{4}\right]$. Then, the conditional distributions $(\mu|A)(\cdot) \in \Delta(\Omega, \mathcal{F})$ and $(\nu|B)(\cdot) \in \Delta(\Omega, \mathcal{F})$ defined by*

$$(\mu|A)(E) := \frac{\mu(A \cap E)}{\mu(A)} \quad \text{and} \quad (\nu|B)(E) := \frac{\nu(B \cap E)}{\nu(B)}$$

*for any event $E \in \mathcal{F}$, satisfy the bound*

$$|\text{TV}(\mu|A, \nu|B) - \text{TV}(\mu, \nu)| \leq 2\epsilon. \tag{29}$$

*Proof of Lemma A.1.* Due to the triangle inequality for the total variation (TV) distance, it follows that

$$\text{TV}(\mu, \nu) \leq \text{TV}(\mu, \mu|A) + \text{TV}(\mu|A, \nu|B) + \text{TV}(\nu|B, \nu), \tag{30}$$

and

$$\text{TV}(\mu|A, \nu|B) \leq \text{TV}(\mu|A, \mu) + \text{TV}(\mu, \nu) + \text{TV}(\nu, \nu|B). \tag{31}$$

At this point, one can easily observe that

$$\begin{aligned}
\text{TV}(\mu, \mu|A) &= \sup\{|\mu(E) - (\mu|A)(E)| : E \in \mathcal{F}\} = (\mu|A)(A) - \mu(A) = 1 - \mu(A); \\
\text{TV}(\nu, \nu|B) &= \sup\{|\nu(E) - (\nu|B)(E)| : E \in \mathcal{F}\} = (\nu|B)(B) - \nu(B) = 1 - \nu(B).
\end{aligned} \tag{32}$$

Putting the observation (32) into the inequalities (30) and (31), the assumptions $1 - \mu(A) \leq \epsilon$ and $1 - \nu(B) \leq \epsilon$ establish the desired result.

$\square$

# B   Proofs and omitted details for Section 3

## B.1   Proof of Proposition 3.1

First, one can observe that

$$\begin{aligned}
&\mathbb{E}_{\mathcal{I}^*}\left[\hat{\tau}_n^f(\mathbf{O}_n)\right] \\
&= \frac{1}{n}\sum_{i=1}^{n}\mathbb{E}_{\mathcal{I}^*}\left[\mathbb{E}_{\mathcal{I}^*}\left[\frac{g(X_i, A_i)Y_i}{\pi_i^*(X_i, \mathbf{O}_{i-1}; A_i)} - f_i(X_i, \mathbf{O}_{i-1}, A_i)\right.\right. \\
&\qquad\qquad \left.\left. + \langle f_i(X_i, \mathbf{O}_{i-1}, \cdot), \pi_i^*(X_i, \mathbf{O}_{i-1}; \cdot)\rangle_{\lambda_\mathbb{A}} \,\Big|\, (X_i, A_i, \mathcal{H}_{i-1})\right]\right] \\
&= \frac{1}{n}\sum_{i=1}^{n}\mathbb{E}_{\mathcal{I}^*}\left[\frac{g(X_i, A_i)\mu^*(X_i, A_i)}{\pi_i^*(X_i, \mathbf{O}_{i-1}; A_i)} - f_i(X_i, \mathbf{O}_{i-1}, A_i) + \langle f_i(X_i, \mathbf{O}_{i-1}, \cdot), \pi_i^*(X_i, \mathbf{O}_{i-1}; \cdot)\rangle_{\lambda_\mathbb{A}}\right] \\
&= \frac{1}{n}\sum_{i=1}^{n}\mathbb{E}_{\mathcal{I}^*}\left[\mathbb{E}_{\mathcal{I}^*}\left[\frac{g(X_i, A_i)\mu^*(X_i, A_i)}{\pi_i^*(X_i, \mathbf{O}_{i-1}; A_i)} - f_i(X_i, \mathbf{O}_{i-1}, A_i)\right.\right. \\
&\qquad\qquad \left.\left. + \langle f_i(X_i, \mathbf{O}_{i-1}, \cdot), \pi_i^*(X_i, \mathbf{O}_{i-1}; \cdot)\rangle_{\lambda_\mathbb{A}} \,\Big|\, (X_i, \mathcal{H}_{i-1})\right]\right] \tag{33} \\
&= \frac{1}{n}\sum_{i=1}^{n}\mathbb{E}_{\mathcal{I}^*}\left[\int_{\mathbb{A}} g(X_i, a)\mu^*(X_i, a)\,\mathrm{d}\lambda_\mathbb{A}(a) - \langle f_i(X_i, \mathbf{O}_{i-1}, \cdot), \pi_i^*(X_i, \mathbf{O}_{i-1}; \cdot)\rangle_{\lambda_\mathbb{A}}\right. \\
&\qquad\qquad \left. + \langle f_i(X_i, \mathbf{O}_{i-1}, \cdot), \pi_i^*(X_i, \mathbf{O}_{i-1}; \cdot)\rangle_{\lambda_\mathbb{A}}\right] \\
&= \tau(\mathcal{I}^*).
\end{aligned}$$

We now assume (3) and note that

$$
\mathrm{Var}_{\mathcal{I}^*}\left[\hat{\tau}_n^f\left(\mathbf{O}_n\right)\right] = \frac{1}{n^2}\sum_{i=1}^n \mathrm{Var}_{\mathcal{I}^*}\left[\frac{g\left(X_i, A_i\right)Y_i}{\pi_i^*\left(X_i, \mathbf{O}_{i-1}; A_i\right)} - f_i\left(X_i, \mathbf{O}_{i-1}, A_i\right)\right]
$$

$$
+ \frac{2}{n^2}\sum_{1 \le i < j \le n}\mathrm{Cov}_{\mathcal{I}^*}\left[\frac{g\left(X_i, A_i\right)Y_i}{\pi_i^*\left(X_i, \mathbf{O}_{i-1}; A_i\right)} - f_i\left(X_i, \mathbf{O}_{i-1}, A_i\right),\right. \tag{34}
$$

$$
\left.\frac{g\left(X_j, A_j\right)Y_j}{\pi_j^*\left(X_j, \mathbf{O}_{j-1}; A_j\right)} - f_j\left(X_j, \mathbf{O}_{j-1}, A_j\right)\right].
$$

One can reveal that

$$
\mathrm{Var}_{\mathcal{I}^*}\left[\frac{g\left(X_i, A_i\right)Y_i}{\pi_i^*\left(X_i, \mathbf{O}_{i-1}; A_i\right)} - f_i\left(X_i, \mathbf{O}_{i-1}, A_i\right)\right]
$$

$$
= \mathbb{E}_{\mathcal{I}^*}\left[\mathbb{E}_{\mathcal{I}^*}\left[\left\{\frac{g\left(X_i, A_i\right)Y_i}{\pi_i^*\left(X_i, \mathbf{O}_{i-1}; A_i\right)} - f_i\left(X_i, \mathbf{O}_{i-1}, A_i\right)\right\}^2\middle|\left(X_i, A_i, \mathcal{H}_{i-1}\right)\right]\right] - \left\{\tau\left(\mathcal{I}^*\right)\right\}^2
$$

$$
= \mathbb{E}_{\mathcal{I}^*}\left[\frac{g^2\left(X_i, A_i\right)}{\left(\pi_i^*\right)^2\left(X_i, \mathbf{O}_{i-1}; A_i\right)}\mathbb{E}_{\mathcal{I}^*}\left[Y_i^2\middle|\left(X_i, A_i, \mathcal{H}_{i-1}\right)\right]\right.
$$

$$
\left. - \frac{2f_i\left(X_i, \mathbf{O}_{i-1}, A_i\right)g\left(X_i, A_i\right)}{\pi_i^*\left(X_i, \mathbf{O}_{i-1}; A_i\right)}\mathbb{E}_{\mathcal{I}^*}\left[Y_i\middle|\left(X_i, A_i, \mathcal{H}_{i-1}\right)\right] + f_i^2\left(X_i, \mathbf{O}_{i-1}, A_i\right)\right] - \left\{\tau\left(\mathcal{I}^*\right)\right\}^2
$$

$$
= \mathbb{E}_{\mathcal{I}^*}\left[\frac{g^2\left(X_i, A_i\right)\sigma^2\left(X_i, A_i\right)}{\left(\pi_i^*\right)^2\left(X_i, \mathbf{O}_{i-1}; A_i\right)}\right]
$$

$$
+ \mathbb{E}_{\mathcal{I}^*}\left[\left\{\frac{g\left(X_i, A_i\right)\mu^*\left(X_i, A_i\right)}{\pi_i^*\left(X_i, \mathbf{O}_{i-1}; A_i\right)} - f_i\left(X_i, \mathbf{O}_{i-1}, A_i\right)\right\}^2\right] \tag{35}
$$

$$
- \left\{\tau\left(\mathcal{I}^*\right)\right\}^2
$$

$$
\stackrel{\text{(a)}}{=} \mathbb{E}_{\mathcal{I}^*}\left[\frac{g^2\left(X_i, A_i\right)\sigma^2\left(X_i, A_i\right)}{\left(\pi_i^*\right)^2\left(X_i, \mathbf{O}_{i-1}; A_i\right)}\right]
$$

$$
+ \mathbb{E}_{\mathcal{I}^*}\left[\left\{\frac{g\left(X_i, A_i\right)\mu^*\left(X_i, A_i\right)}{\pi_i^*\left(X_i, \mathbf{O}_{i-1}; A_i\right)} - \left\langle g\left(X_i, \cdot\right), \mu^*\left(X_i, \cdot\right)\right\rangle_{\lambda_{\mathbb{A}}} - f_i\left(X_i, \mathbf{O}_{i-1}, A_i\right)\right\}^2\right]
$$

$$
+ \underbrace{\mathbb{E}_{\mathcal{I}^*}\left[\left\langle g\left(X_i, \cdot\right), \mu^*\left(X_i, \cdot\right)\right\rangle_\lambda^2\right] - \left\{\tau\left(\mathcal{I}^*\right)\right\}^2}_{= \mathrm{Var}_{X \sim \Xi^*}\left[\left\langle g(X, \cdot), \mu^*(X, \cdot)\right\rangle_{\lambda_{\mathbb{A}}}\right]}
$$

$$
= \mathbb{E}_{\mathcal{I}^*}\left[\frac{g^2\left(X_i, A_i\right)\sigma^2\left(X_i, A_i\right)}{\left(\pi_i^*\right)^2\left(X_i, \mathbf{O}_{i-1}; A_i\right)}\right]
$$

$$
+ \mathbb{E}_{\mathcal{I}^*}\left[\left\{\frac{g\left(X_i, A_i\right)\mu^*\left(X_i, A_i\right)}{\pi_i^*\left(X_i, \mathbf{O}_{i-1}; A_i\right)} - \left\langle g\left(X_i, \cdot\right), \mu^*\left(X_i, \cdot\right)\right\rangle_{\lambda_{\mathbb{A}}} - f_i\left(X_i, \mathbf{O}_{i-1}, A_i\right)\right\}^2\right]
$$

$$
+ \mathrm{Var}_{X \sim \Xi^*}\left[\left\langle g\left(X, \cdot\right), \mu^*\left(X, \cdot\right)\right\rangle_{\lambda_{\mathbb{A}}}\right],
$$

where the step (a) can be verified as follows:

$$
\mathbb{E}_{\mathcal{I}^*}\left[\left\{\frac{g\left(X_i, A_i\right)\mu^*\left(X_i, A_i\right)}{\pi_i^*\left(X_i, \mathbf{O}_{i-1}; A_i\right)} - f_i\left(X_i, \mathbf{O}_{i-1}, A_i\right)\right\}^2\right]
$$

$$
= \mathbb{E}_{\mathcal{I}^*}\left[\mathbb{E}_{\mathcal{I}^*}\left[\left\{\frac{g\left(X_i, A_i\right)\mu^*\left(X_i, A_i\right)}{\pi_i^*\left(X_i, \mathbf{O}_{i-1}; A_i\right)} - f_i\left(X_i, \mathbf{O}_{i-1}, A_i\right)\right\}^2\middle|\left(X_i, \mathcal{H}_{i-1}\right)\right]\right]
$$

$$
= \mathbb{E}_{\mathcal{I}^*}\left[\mathrm{Var}_{\mathcal{I}^*}\left[\frac{g\left(X_i, A_i\right)\mu^*\left(X_i, A_i\right)}{\pi_i^*\left(X_i, \mathbf{O}_{i-1}; A_i\right)} - f_i\left(X_i, \mathbf{O}_{i-1}, A_i\right)\middle|\left(X_i, \mathcal{H}_{i-1}\right)\right]\right]
$$

$$+ \mathbb{E}_{\mathcal{I}^*} \left[ \left( \underbrace{\mathbb{E}_{\mathcal{I}^*} \left[ \frac{g\left(X_i, A_i\right) \mu^*\left(X_i, A_i\right)}{\pi_i^*\left(X_i, \mathbf{O}_{i-1}; A_i\right)} - f_i\left(X_i, \mathbf{O}_{i-1}, A_i\right) \middle| \left(X_i, \mathcal{H}_{i-1}\right) \right]}_{= \langle g(X_i, \cdot), \mu^*(X_i, \cdot) \rangle_\lambda} \right)^2 \right]$$

$$= \mathbb{E}_{\mathcal{I}^*} \left[ \left\{ \frac{g\left(X_i, A_i\right) \mu^*\left(X_i, A_i\right)}{\pi_i^*\left(X_i, \mathbf{O}_{i-1}; A_i\right)} - \langle g\left(X_i, \cdot\right), \mu^*\left(X_i, \cdot\right) \rangle_{\lambda_{\mathbb{A}}} - f_i\left(X_i, \mathbf{O}_{i-1}, A_i\right) \right\}^2 \right]$$

$$+ \mathbb{E}_{\mathcal{I}^*} \left[ \langle g\left(X_i, \cdot\right), \mu^*\left(X_i, \cdot\right) \rangle_{\lambda_{\mathbb{A}}}^2 \right].$$

Next, we compute $\text{Cov}_{\mathcal{I}^*} \left[ \frac{g(X_i, A_i) Y_i}{\pi_i^*(X_i, \mathbf{O}_{i-1}; A_i)} - f_i\left(X_i, \mathbf{O}_{i-1}, A_i\right), \frac{g(X_j, A_j) Y_j}{\pi_j^*(X_j, \mathbf{O}_{j-1}; A_j)} - f_j\left(X_j, \mathbf{O}_{j-1}, A_j\right) \right]$:

$$\text{Cov}_{\mathcal{I}^*} \left[ \frac{g\left(X_i, A_i\right) Y_i}{\pi_i^*\left(X_i, \mathbf{O}_{i-1}; A_i\right)} - f_i\left(X_i, \mathbf{O}_{i-1}, A_i\right), \frac{g\left(X_j, A_j\right) Y_j}{\pi_j^*\left(X_j, \mathbf{O}_{j-1}; A_j\right)} - f_j\left(X_j, \mathbf{O}_{j-1}, A_j\right) \right]$$

$$= \mathbb{E}_{\mathcal{I}^*} \left[ \left\{ \frac{g\left(X_i, A_i\right) Y_i}{\pi_i^*\left(X_i, \mathbf{O}_{i-1}; A_i\right)} - f_i\left(X_i, \mathbf{O}_{i-1}, A_i\right) \right\} \right.$$

$$\left. \left\{ \frac{g\left(X_j, A_j\right) \mu^*\left(X_j, A_j\right)}{\pi_j^*\left(X_j, \mathbf{O}_{j-1}; A_j\right)} - f_j\left(X_j, \mathbf{O}_{j-1}, A_j\right) \right\} \right] - \left\{ \tau\left(\mathcal{I}^*\right) \right\}^2$$

$$= \mathbb{E}_{\mathcal{I}^*} \left[ \mathbb{E}_{\mathcal{I}^*} \left[ \left\{ \frac{g\left(X_i, A_i\right) Y_i}{\pi_i^*\left(X_i, \mathbf{O}_{i-1}; A_i\right)} - f_i\left(X_i, \mathbf{O}_{i-1}, A_i\right) \right\} \right. \right. \tag{36}$$

$$\left. \left. \left\{ \frac{g\left(X_j, A_j\right) \mu^*\left(X_j, A_j\right)}{\pi_j^*\left(X_j, \mathbf{O}_{j-1}; A_j\right)} - f_j\left(X_j, \mathbf{O}_{j-1}, A_j\right) \right\} \middle| \left(X_j, \mathcal{H}_{j-1}\right) \right] \right] - \left\{ \tau\left(\mathcal{I}^*\right) \right\}^2$$

$$= \mathbb{E}_{\mathcal{I}^*} \left[ \left\{ \frac{g\left(X_i, A_i\right) Y_i}{\pi_i^*\left(X_i, \mathbf{O}_{i-1}; A_i\right)} - f_i\left(X_i, \mathbf{O}_{i-1}, A_i\right) \right\} \langle g\left(X_j, \cdot\right), \mu^*\left(X_j, \cdot\right) \rangle_{\lambda_{\mathbb{A}}} \right] - \left\{ \tau\left(\mathcal{I}^*\right) \right\}^2$$

$$= \mathbb{E}_{\mathcal{I}^*} \left[ \mathbb{E}_{\mathcal{I}^*} \left[ \left\{ \frac{g\left(X_i, A_i\right) Y_i}{\pi_i^*\left(X_i, \mathbf{O}_{i-1}; A_i\right)} - f_i\left(X_i, \mathbf{O}_{i-1}, A_i\right) \right\} \langle g\left(X_j, \cdot\right), \mu^*\left(X_j, \cdot\right) \rangle_{\lambda_{\mathbb{A}}} \middle| \mathcal{H}_{j-1} \right] \right] - \left\{ \tau\left(\mathcal{I}^*\right) \right\}^2$$

$$\overset{(b)}{=} 0,$$

where the step (b) holds due to the fact that $X_j$ is independent of the historical data $\mathcal{H}_{j-1}$, which immediately yields $X_j | \mathcal{H}_{j-1} \overset{d}{=} X_j \sim \Xi^*(\cdot)$. Taking two pieces (35) and (36) collectively into the equation (34), one has

$$n \cdot \text{Var}_{\mathcal{I}^*} \left[ \hat{\tau}_n^f\left(\mathbf{O}_n\right) \right]$$

$$= \text{Var}_{X \sim \Xi^*} \left[ \langle g(X, \cdot), \mu^*(X, \cdot) \rangle_{\lambda_{\mathbb{A}}} \right] + \frac{1}{n} \sum_{i=1}^n \left( \mathbb{E}_{\mathcal{I}^*} \left[ \frac{g^2\left(X_i, A_i\right) \sigma^2\left(X_i, A_i\right)}{\left(\pi_i^*\right)^2\left(X_i, \mathbf{O}_{i-1}, A_i\right)} \right] \right.$$

$$\left. + \mathbb{E}_{\mathcal{I}^*} \left[ \left\{ \frac{g\left(X_i, A_i\right) \mu^*\left(X_i, A_i\right)}{\pi_i^*\left(X_i, \mathbf{O}_{i-1}; A_i\right)} - \langle g\left(X_i, \cdot\right), \mu^*\left(X_i, \cdot\right) \rangle_{\lambda_{\mathbb{A}}} - f_i\left(X_i, \mathbf{O}_{i-1}, A_i\right) \right\}^2 \right] \right),$$

as desired.

## B.2 Proof of Theorem 3.1

We first single out a key technical lemma throughout this section that plays a crucial role in the proof of Theorem 3.1.

**Lemma B.1.** *The following results hold:*

(i) It holds that $\mathbb{E}_{\mathcal{I}^*}\left[\hat{\Gamma}_i\left(\mathbf{O}_i\right)\Big|\left(X_i, \mathcal{H}_{i-1}\right)\right] = \left\langle g\left(X_i, \cdot\right), \mu^*\left(X_i, \cdot\right)\right\rangle_{\lambda_{\mathbb{A}}}$ *for all* $i \in [n]$. *Therefore, one has*

$$
\begin{aligned}
\mathbb{E}_{\mathcal{I}^*}\left[\hat{\Gamma}_i\left(\mathbf{O}_i\right)\right] &= \mathbb{E}_{\mathcal{I}^*}\left[\mathbb{E}_{\mathcal{I}^*}\left[\hat{\Gamma}_i\left(\mathbf{O}_i\right)\Big|\left(X_i, \mathcal{H}_{i-1}\right)\right]\right] \\
&= \mathbb{E}_{\mathcal{I}^*}\left[\left\langle g\left(X_i, \cdot\right), \mu^*\left(X_i, \cdot\right)\right\rangle_{\lambda_{\mathbb{A}}}\right] \\
&= \tau\left(\mathcal{I}^*\right).
\end{aligned}
\tag{37}
$$

(ii) *For every* $1 \leq i < j \leq n$, *we have* $\mathrm{Cov}_{\mathcal{I}^*}\left[\hat{\Gamma}_i\left(\mathbf{O}_i\right), \hat{\Gamma}_j\left(\mathbf{O}_j\right)\right] = 0$;

(iii) *For every* $i \in [n]$,

$$
\begin{aligned}
&\mathrm{Var}_{\mathcal{I}^*}\left[\hat{\Gamma}_i\left(\mathbf{O}_i\right)\right] \\
&= \mathrm{Var}_{X \sim \Xi^*}\left[\left\langle g\left(X, \cdot\right), \mu^*\left(X, \cdot\right)\right\rangle_{\lambda_{\mathbb{A}}}\right] + \mathbb{E}_{\mathcal{I}^*}\left[\frac{g^2\left(X_i, A_i\right)\sigma^2\left(X_i, A_i\right)}{\left(\pi_i^*\right)^2\left(X_i, \mathbf{O}_{i-1}; A_i\right)}\right] \\
&\quad + \mathbb{E}_{\mathcal{I}^*}\left[\mathrm{Var}_{\mathcal{I}^*}\left[\frac{g\left(X_i, A_i\right)}{\pi_i^*\left(X_i, \mathbf{O}_{i-1}; A_i\right)}\left\{\hat{\mu}_i\left(\mathbf{O}_{i-1}\right)\left(X_i, A_i\right) - \mu^*\left(X_i, A_i\right)\right\}\Big|\left(X_i, \mathcal{H}_{i-1}\right)\right]\right] \\
&\leq \mathrm{Var}_{X \sim \Xi^*}\left[\left\langle g\left(X, \cdot\right), \mu^*\left(X, \cdot\right)\right\rangle_{\lambda_{\mathbb{A}}}\right] + \mathbb{E}_{\mathcal{I}^*}\left[\frac{g^2\left(X_i, A_i\right)\sigma^2\left(X_i, A_i\right)}{\left(\pi_i^*\right)^2\left(X_i, \mathbf{O}_{i-1}; A_i\right)}\right] \\
&\quad + \mathbb{E}_{\mathcal{I}^*}\left[\frac{g^2\left(X_i, A_i\right)\left\{\hat{\mu}_i\left(\mathbf{O}_{i-1}\right)\left(X_i, A_i\right) - \mu^*\left(X_i, A_i\right)\right\}^2}{\left(\pi_i^*\right)^2\left(X_i, \mathbf{O}_{i-1}; A_i\right)}\right].
\end{aligned}
\tag{38}
$$

*Proof of Lemma B.1.*
(i) From the definition of $\hat{\Gamma}_i(\cdot) : \mathbb{O}^i \to \mathbb{R}$ in (9), we have

$$
\begin{aligned}
\mathbb{E}_{\mathcal{I}^*}\left[\hat{\Gamma}_i\left(\mathbf{O}_i\right)\Big|\left(X_i, A_i, \mathcal{H}_{i-1}\right)\right] &= \frac{g\left(X_i, A_i\right)}{\pi_i^*\left(X_i, \mathbf{O}_{i-1}; A_i\right)}\left\{\mu^*\left(X_i, A_i\right) - \hat{\mu}_i\left(\mathbf{O}_{i-1}\right)\left(X_i, A_i\right)\right\} \\
&\quad + \left\langle g\left(X_i, \cdot\right), \hat{\mu}_i\left(\mathbf{O}_{i-1}\right)\left(X_i, \cdot\right)\right\rangle_{\lambda_{\mathbb{A}}}.
\end{aligned}
\tag{39}
$$

Thus, we obtain

$$
\begin{aligned}
&\mathbb{E}_{\mathcal{I}^*}\left[\hat{\Gamma}_i\left(\mathbf{O}_i\right)\Big|\left(X_i, \mathcal{H}_{i-1}\right)\right] \\
&= \mathbb{E}_{\mathcal{I}^*}\left[\mathbb{E}_{\mathcal{I}^*}\left[\hat{\Gamma}_i\left(\mathbf{O}_i\right)\Big|\left(X_i, A_i, \mathcal{H}_{i-1}\right)\right]\Big|\left(X_i, \mathcal{H}_{i-1}\right)\right] \\
&= \int_{\mathbb{A}}\frac{g\left(X_i, a\right)}{\pi_i^*\left(X_i, \mathbf{O}_{i-1}; a\right)}\left\{\mu^*\left(X_i, a\right) - \hat{\mu}_i\left(\mathbf{O}_{i-1}\right)\left(X_i, a\right)\right\} \cdot \pi_i^*\left(X_i, \mathbf{O}_{i-1}; a\right)\mathrm{d}\lambda_{\mathbb{A}}(a) \\
&\quad + \left\langle g\left(X_i, \cdot\right), \hat{\mu}_i\left(\mathbf{O}_{i-1}\right)\left(X_i, \cdot\right)\right\rangle_{\lambda_{\mathbb{A}}} \\
&= \left\langle g\left(X_i, \cdot\right), \mu^*\left(X_i, \cdot\right)\right\rangle_{\lambda_{\mathbb{A}}}
\end{aligned}
\tag{40}
$$

as desired.

(ii) One can reveal that

$$
\begin{aligned}
&\mathrm{Cov}_{\mathcal{I}^*}\left[\hat{\Gamma}_i\left(\mathbf{O}_i\right), \hat{\Gamma}_j\left(\mathbf{O}_j\right)\right] \\
&= \mathbb{E}_{\mathcal{I}^*}\left[\hat{\Gamma}_i\left(\mathbf{O}_i\right)\mathbb{E}\left[\hat{\Gamma}_j\left(\mathbf{O}_j\right)\Big|\left(X_j, A_j, \mathcal{H}_{j-1}\right)\right]\right] - \left\{\tau\left(\mathcal{I}^*\right)\right\}^2 \\
&= \mathbb{E}_{\mathcal{I}^*}\left[\hat{\Gamma}_i\left(\mathbf{O}_i\right)\left[\frac{g\left(X_j, A_j\right)}{\pi_j^*\left(X_j, \mathbf{O}_{j-1}; A_j\right)}\left\{\mu^*\left(X_j, A_j\right) - \hat{\mu}_j\left(\mathbf{O}_{j-1}\right)\left(X_j, A_j\right)\right\}\right.\right. \\
&\quad \left.\left. + \left\langle g\left(X_j, \cdot\right), \hat{\mu}_j\left(\mathbf{O}_{j-1}\right)\left(X_j, \cdot\right)\right\rangle_{\lambda_{\mathbb{A}}}\right]\right] - \left\{\tau\left(\mathcal{I}^*\right)\right\}^2
\end{aligned}
\tag{41}
$$

$$= \mathbb{E}_{\mathcal{I}^*} \left[ \hat{\Gamma}_i(\mathbf{O}_i) \, \mathbb{E}_{\mathcal{I}^*} \left[ \frac{g(X_j, A_j)}{\pi_j^*(X_j, \mathbf{O}_{j-1}; A_j)} \left\{ \mu^*(X_j, A_j) - \hat{\mu}_j(\mathbf{O}_{j-1})(X_j, A_j) \right\} \right. \right.$$

$$\left. \left. + \left\langle g(X_j, \cdot), \hat{\mu}_j(\mathbf{O}_{j-1})(X_j, \cdot) \right\rangle_{\lambda_{\mathbb{A}}} \middle| (X_j, \mathcal{H}_{j-1}) \right] \right] - \left\{ \tau(\mathcal{I}^*) \right\}^2$$

$$= \mathbb{E}_{\mathcal{I}^*} \left[ \hat{\Gamma}_i(\mathbf{O}_i; g) \left\langle g(X_j, \cdot), \mu^*(X_j, \cdot) \right\rangle_{\lambda_{\mathbb{A}}} \right] - \left\{ \tau(\mathcal{I}^*; g) \right\}^2$$

$$\overset{(a)}{=} 0,$$

where the step (a) holds due to the facts that $\hat{\Gamma}_i(\mathbf{O}_i)$ is $\mathcal{H}_{j-1}$-measurable and $X_j \perp\!\!\!\perp \mathcal{H}_{j-1}$, together with the equation (37).

(iii) It follows that

$$\text{Var}_{\mathcal{I}^*} \left[ \hat{\Gamma}_i(\mathbf{O}_i) \right]$$

$$= \mathbb{E}_{\mathcal{I}^*} \left[ \text{Var}_{\mathcal{I}^*} \left[ \hat{\Gamma}_i(\mathbf{O}_i) \middle| (X_i, \mathcal{H}_{i-1}) \right] \right] + \text{Var}_{\mathcal{I}^*} \left[ \mathbb{E}_{\mathcal{I}^*} \left[ \hat{\Gamma}_i(\mathbf{O}_i) \middle| (X_i, \mathcal{H}_{i-1}) \right] \right]$$

$$\overset{(b)}{=} \mathbb{E}_{\mathcal{I}^*} \left[ \mathbb{E}_{\mathcal{I}^*} \left[ \text{Var}_{\mathcal{I}^*} \left[ \hat{\Gamma}_i(\mathbf{O}_i) \middle| (X_i, A_i, \mathcal{H}_{i-1}) \right] \middle| (X_i, \mathcal{H}_{i-1}) \right] \right]$$

$$+ \mathbb{E}_{\mathcal{I}^*} \left[ \text{Var}_{\mathcal{I}^*} \left[ \mathbb{E}_{\mathcal{I}^*} \left[ \hat{\Gamma}_i(\mathbf{O}_i) \middle| (X_i, A_i, \mathcal{H}_{i-1}) \right] \middle| (X_i, \mathcal{H}_{i-1}) \right] \right] \tag{42}$$

$$+ \text{Var}_{X \sim \Xi^*} \left[ \left\langle g(X, \cdot), \mu^*(X, \cdot) \right\rangle_{\lambda_{\mathbb{A}}} \right]$$

$$= \mathbb{E}_{\mathcal{I}^*} \left[ \frac{g^2(X_i, A_i) \, \sigma^2(X_i, A_i)}{(\pi_i^*)^2(X_i, \mathbf{O}_{i-1}; A_i)} \right]$$

$$+ \mathbb{E}_{\mathcal{I}^*} \left[ \text{Var}_{\mathcal{I}^*} \left[ \frac{g(X_i, A_i)}{\pi_i^*(X_i, \mathbf{O}_{i-1}; A_i)} \left\{ \mu^*(X_i, A_i) - \hat{\mu}_i(\mathbf{O}_{i-1})(X_i, A_i) \right\} \middle| (X_i, \mathcal{H}_{i-1}) \right] \right]$$

$$+ \text{Var}_{X \sim \Xi^*} \left[ \left\langle g(X, \cdot), \mu^*(X, \cdot) \right\rangle_{\lambda_{\mathbb{A}}} \right],$$

as desired, where the step (b) follows from the fact (40).

$\square$

Now, it's time to finish the proof of Theorem 3.1. One can reveal that

$$\mathbb{E}_{\mathcal{I}^*} \left[ \left\{ \hat{\tau}_n^{\text{AIPW}}(\mathbf{O}_n; g) - \tau(\mathcal{I}^*; g) \right\}^2 \right]$$

$$\overset{(a)}{=} \frac{1}{n^2} \sum_{i=1}^n \text{Var}_{\mathcal{I}^*} \left[ \hat{\Gamma}_i(\mathbf{O}_i; g) \right]$$

$$\overset{(b)}{\leq} \frac{1}{n^2} \sum_{i=1}^n \left\{ \text{Var}_{X \sim \Xi^*} \left[ \left\langle g(X, \cdot), \mu^*(X, \cdot) \right\rangle_{\lambda_{\mathbb{A}}} \right] + \mathbb{E}_{\mathcal{I}^*} \left[ \frac{g^2(X_i, A_i) \, \sigma^2(X_i, A_i)}{(\pi_i^*)^2(X_i, \mathbf{O}_{i-1}; A_i)} \right] \right.$$

$$\left. + \mathbb{E}_{\mathcal{I}^*} \left[ \frac{g^2(X_i, A_i) \left\{ \mu^*(X_i, A_i) - \hat{\mu}_i(\mathbf{O}_{i-1})(X_i, A_i) \right\}^2}{(\pi_i^*)^2(X_i, \mathbf{O}_{i-1}; A_i)} \right] \right\}$$

$$\overset{(c)}{=} \frac{1}{n} \left\{ v_*^2 + \frac{1}{n} \sum_{i=1}^n \mathbb{E}_{\mathcal{I}^*} \left[ \frac{g^2(X_i, A_i) \left\{ \hat{\mu}_i(\mathbf{O}_{i-1})(X_i, A_i) - \mu^*(X_i, A_i) \right\}^2}{(\pi_i^*)^2(X_i, \mathbf{O}_{i-1}; A_i)} \right] \right\},$$

where the step (a) holds from the part (ii) of Lemma B.1, the step (b) makes use of the inequality (38), and the step (c) follows from the definition of $v_*^2$ in (7).

## B.3 Proof of Theorem 3.2

It holds due to the observation (12) that

$$\mathbb{E}_{\mathcal{I}^*} \left[ \sum_{i=1}^n l_i \left\{ \hat{\mu}_i(\mathbf{O}_{i-1}) \right\} \right]$$

$$= \sum_{i=1}^{n} \mathbb{E}_{\mathcal{I}^*} \left[ \mathbb{E}_{\mathcal{I}^*} \left[ l_i \left\{ \hat{\mu}_i \left( \mathbf{O}_{i-1} \right) \right\} | \left( \mathcal{H}_{i-1}, X_i, A_i \right) \right] \right]$$

$$= \sum_{i=1}^{n} \mathbb{E}_{\mathcal{I}^*} \left[ \frac{g^2 \left( X_i, A_i \right)}{\left( \pi_i^* \right)^2 \left( X_i, \mathbf{O}_{i-1}; A_i \right)} \left[ \sigma^2 \left( X_i, A_i \right) + \left\{ \hat{\mu}_i \left( \mathbf{O}_{i-1} \right) \left( X_i, A_i \right) - \mu^* \left( X_i, A_i \right) \right\}^2 \right] \right]$$

$$= n \left\| \sigma \right\|_{(n)}^2 + \sum_{i=1}^{n} \mathbb{E}_{\mathcal{I}^*} \left[ \frac{g^2 \left( X_i, A_i \right) \left\{ \hat{\mu}_i \left( \mathbf{O}_{i-1} \right) \left( X_i, A_i \right) - \mu^* \left( X_i, A_i \right) \right\}^2}{\left( \pi_i^* \right)^2 \left( X_i, \mathbf{O}_{i-1}; A_i \right)} \right],$$

which establishes the following expression of the estimation error term (11):

$$\frac{1}{n} \sum_{i=1}^{n} \mathbb{E}_{\mathcal{I}^*} \left[ \frac{g^2 \left( X_i, A_i \right) \left\{ \hat{\mu}_i \left( \mathbf{O}_{i-1} \right) \left( X_i, A_i \right) - \mu^* \left( X_i, A_i \right) \right\}^2}{\left( \pi_i^* \right)^2 \left( X_i, \mathbf{O}_{i-1}; A_i \right)} \right]$$

$$= \frac{1}{n} \mathbb{E}_{\mathcal{I}^*} \left[ \sum_{i=1}^{n} l_i \left\{ \hat{\mu}_i \left( \mathbf{O}_{i-1} \right) \right\} \right] - \left\| \sigma \right\|_{(n)}^2 \tag{43}$$

$$= \frac{1}{n} \mathbb{E}_{\mathcal{I}^*} \left[ \text{Regret} \left( n; \mathcal{A} \right) \right] + \frac{1}{n} \mathbb{E}_{\mathcal{I}^*} \left[ \inf \left\{ \sum_{i=1}^{n} l_i(\mu) : \mu \in \mathcal{F} \right\} \right] - \left\| \sigma \right\|_{(n)}^2.$$

At this point, one can realize that

$$\frac{1}{n} \mathbb{E}_{\mathcal{I}^*} \left[ \inf \left\{ \sum_{i=1}^{n} l_i(\mu) : \mu \in \mathcal{F} \right\} \right]$$

$$\leq \inf \left\{ \frac{1}{n} \mathbb{E}_{\mathcal{I}^*} \left[ \sum_{i=1}^{n} l_i(\mu) \right] : \mu \in \mathcal{F} \right\}$$

$$= \inf \left\{ \frac{1}{n} \sum_{i=1}^{n} \mathbb{E}_{\mathcal{I}^*} \left[ \mathbb{E}_{\mathcal{I}^*} \left[ l_i(\mu) | \left( \mathcal{H}_{i-1}, X_i, A_i \right) \right] \right] : \mu \in \mathcal{F} \right\} \tag{44}$$

$$\stackrel{(a)}{=} \inf \left\{ \frac{1}{n} \sum_{i=1}^{n} \mathbb{E}_{\mathcal{I}^*} \left[ \frac{g^2 \left( X_i, A_i \right)}{\left( \pi_i^* \right)^2 \left( X_i, \mathbf{O}_{i-1}; A_i \right)} \left[ \sigma^2 \left( X_i, A_i \right) + \left\{ \mu \left( X_i, A_i \right) - \mu^* \left( X_i, A_i \right) \right\}^2 \right] \right] : \mu \in \mathcal{F} \right\}$$

$$= \left\| \sigma \right\|_{(n)}^2 + \inf \left\{ \left\| \mu - \mu^* \right\|_{(n)}^2 : \mu \in \mathcal{F} \right\},$$

where the step (a) holds by the fact (12). Taking two pieces (43) and (44) collectively, it follows that

$$\frac{1}{n} \sum_{i=1}^{n} \mathbb{E}_{\mathcal{I}^*} \left[ \frac{g^2 \left( X_i, A_i \right) \left\{ \hat{\mu}_i \left( \mathbf{O}_{i-1} \right) \left( X_i, A_i \right) - \mu^* \left( X_i, A_i \right) \right\}^2}{\left( \pi_i^* \right)^2 \left( X_i, \mathbf{O}_{i-1}; A_i \right)} \right]$$

$$\leq \frac{1}{n} \mathbb{E}_{\mathcal{I}^*} \left[ \text{Regret} \left( n; \mathcal{A} \right) \right] + \inf \left\{ \left\| \mu - \mu^* \right\|_{(n)}^2 : \mu \in \mathcal{F} \right\}. \tag{45}$$

Hence, the upper bound (15) on the MSE of the AIPW estimator (8) is an immediate consequence of the inequality (45) by putting it into the bound (10) in Theorem 3.1.

### B.4 Proof of Theorem 3.3

One can easily observe from the equation (17) for every $\mu \in \mathcal{F}$ that

$$\left\| \nabla l_i(\mu) \right\|_2^2 = \frac{4 g^4 \left( X_i, A_i \right)}{\left( \pi_i^* \right)^4 \left( X_i, \mathbf{O}_{i-1}; A_i \right)} \left\{ Y_i - \mu \left( X_i, A_i \right) \right\}^2 \stackrel{\mathbb{P}_{\mathcal{I}^*}^n\text{-a.s.}}{\leq} \left( 4 L B^2 \right)^2, \tag{46}$$

which holds due to Assumption 1 together with the fact $\mathbb{Y} = [-L, L]$. So it turns out that the loss function (14) is Lipschitz continuous with parameter $G := 4 L B^2$ $\mathbb{P}_{\mathcal{I}^*}^n$-almost surely. Hence, the desired conclusion immediately follows by *Theorem 3.1* in [18] with parameter $G = 4 L B^2$.

## B.5 Proof of Theorem 3.4

One can realize from the equation (22) that $\mathbb{P}_{\mathcal{I}^*}^n$-almost surely,

$$
\begin{aligned}
\|\nabla_{\boldsymbol{\theta}} \mathcal{L}_i(\boldsymbol{\theta})\|_2^2 &= \frac{4g^4(X_i, A_i)}{(\pi_i^*)^4(X_i, \mathbf{O}_{i-1}; A_i)} \left\{\boldsymbol{\theta}^\top \phi(X_i, A_i) - Y_i\right\}^2 \|\phi(X_i, A_i)\|_2^2 \\
&\leq 4B^4 \left\{|Y_i| + \|\boldsymbol{\theta}\|_2 \|\phi(X_i, A_i)\|_2\right\}^2 \|\phi(X_i, A_i)\|_2^2 \\
&\leq 4B^4(L+R)^2,
\end{aligned}
\tag{47}
$$

which holds by Assumption 1 together with the facts $\mathbb{Y} = [-L, L]$ and $\sup_{(x,a) \in \mathbb{X} \times \mathbb{A}} \|\phi(x,a)\|_2 \leq 1$. So, the loss function (21) is Lipschitz continuous with parameter $G := 2B^2(L+R) \ \mathbb{P}_{\mathcal{I}^*}^n$-a.s. Hence, the desired result follows by *Theorem 3.1* in [18] with parameter $G = 2B^2(L+R)$ and $D = 2R$.

## B.6 Consequences for particular outcome models: general function approximation

Lastly, we consider the most challenging setting where the estimation of the treatment effect $\mu^*(\cdot, \cdot) : \mathbb{X} \times \mathbb{A} \to \mathbb{R}$ is parameterized by general function classes. Under Assumption 1, one first observes from the MSE bound (10) of the AIPW estimator (8) in Theorem 3.1 that

$$
\begin{aligned}
&\mathbb{E}_{\mathcal{I}^*} \left[\left\{\hat{\tau}_n^{\mathsf{AIPW}}(\mathbf{O}_n) - \tau(\mathcal{I}^*)\right\}^2\right] \\
&\leq \frac{1}{n} \left\{v_*^2 + \frac{1}{n} \sum_{i=1}^n \mathbb{E}\left[\frac{g^2(X_i, A_i)\left\{\hat{\mu}_i(\mathbf{O}_{i-1})(X_i, A_i) - \mu^*(X_i, A_i)\right\}^2}{(\pi_i^*)^2(X_i, \mathbf{O}_{i-1}; A_i)}\right]\right\} \\
&\leq \frac{1}{n} \left\{v_*^2 + \frac{B^2}{n} \sum_{i=1}^n \mathbb{E}\left[\left\{\hat{\mu}_i(\mathbf{O}_{i-1})(X_i, A_i) - \mu^*(X_i, A_i)\right\}^2\right]\right\}.
\end{aligned}
\tag{48}
$$

From the last term in the MSE bound (48), our aim becomes to control an upper bound of the term

$$
\frac{1}{n} \sum_{i=1}^n \mathbb{E}\left[\left\{\hat{\mu}_i(\mathbf{O}_{i-1})(X_i, A_i) - \mu^*(X_i, A_i)\right\}^2\right]
\tag{49}
$$

in the finite sample regime. Towards achieving this goal, we consider the online non-parametric regression problem described in Algorithm 2 whose sequence $\{l_i(\cdot) : (\mathbb{X} \times \mathbb{A} \to \mathbb{R}) \to \mathbb{R} : i \in [n]\}$ of loss functions defined as (14) is superseded by $\{\bar{l}_i(\cdot) : (\mathbb{X} \times \mathbb{A} \to \mathbb{R}) \to \mathbb{R} : i \in [n]\}$, where

$$
\bar{l}_i(\mu) := \{Y_i - \mu(X_i, A_i)\}^2, \ \forall (\mu, i) \in (\mathbb{X} \times \mathbb{A} \to \mathbb{R}) \times [n].
\tag{50}
$$

It is straightforward to see for every $i \in [n]$ that

$$
\mathbb{E}_{\mathcal{I}^*}\left[\bar{l}_i(\mu) \,\middle|\, (\mathcal{H}_{i-1}, X_i, A_i)\right] = \sigma^2(X_i, A_i) + \{\mu(X_i, A_i) - \mu^*(X_i, A_i)\}^2.
\tag{51}
$$

With this modified online non-parametric regression problem, we now aim to minimize the learner's *modified regret* defined as follows:

$$
\overline{\mathrm{Regret}}(n, \mathcal{F}; \overline{\mathcal{A}}) := \sum_{i=1}^n \bar{l}_i\{\hat{\mu}_i(\mathbf{O}_{i-1})\} - \inf\left\{\sum_{i=1}^n \bar{l}_i(\mu) : \mu \in \mathcal{F}\right\},
\tag{52}
$$

where $\overline{\mathcal{A}}$ denotes the learner's online non-parametric regression algorithm that returns a sequence of estimates $\{\hat{\mu}_i(\mathbf{O}_{i-1}) \in (\mathbb{X} \times \mathbb{A} \to \mathbb{R}) : i \in [n]\}$ of the treatment effect based on interactions with the environment which selects modified loss functions $\{\bar{l}_i(\cdot) : (\mathbb{X} \times \mathbb{A} \to \mathbb{R}) \to \mathbb{R} : i \in [n]\}$.

**Theorem B.1.** *The AIPW estimator* (8) *based on a sequence* $\{\hat{\mu}_i(\mathbf{O}_{i-1}) \in (\mathbb{X} \times \mathbb{A} \to \mathbb{R}) : i \in [n]\}$ *of estimates for the treatment effect* $\mu^*$ *produced by making use of an online non-parametric regression algorithm* $\overline{\mathcal{A}}$ *against the environment which chooses the sequence of modified loss functions*

$\left\{\bar{l}_i(\cdot) : (\mathbb{X} \times \mathbb{A} \to \mathbb{R}) \to \mathbb{R} : i \in [n]\right\}$ *defined in* (50) *enjoys the following upper bound on the MSE:*

$$\mathbb{E}_{\mathcal{I}^*}\left[\left\{\hat{\tau}_n^{\mathsf{AIPW}}(\mathbf{O}_n) - \tau(\mathcal{I}^*)\right\}^2\right]$$

$$\leq \frac{1}{n}\left( v_*^2 + \frac{1}{n}\mathbb{E}_{\mathcal{I}^*}\left[\overline{\text{Regret}}(n, \mathcal{F}; \overline{\mathcal{A}})\right]\right.$$

$$\left. + \underbrace{\inf\left\{\frac{1}{n}\sum_{i=1}^{n}\mathbb{E}_{\mathcal{I}^*}\left[\{\mu(X_i, A_i) - \mu^*(X_i, A_i)\}^2\right] : \mu \in \mathcal{F}\right\}}_{\text{approximation error term.}}\right). \tag{53}$$

*Proof of Theorem B.1.* It follows from the property (51) that

$$\mathbb{E}_{\mathcal{I}^*}\left[\sum_{i=1}^{n}\bar{l}_i\{\hat{\mu}_i(\mathbf{O}_{i-1})\}\right]$$

$$= \sum_{i=1}^{n}\mathbb{E}_{\mathcal{I}^*}\left[\mathbb{E}_{\mathcal{I}^*}\left[\bar{l}_i\{\hat{\mu}_i(\mathbf{O}_{i-1})\}\middle|(\mathcal{F}_{i-1}, X_i, A_i)\right]\right]$$

$$= \sum_{i=1}^{n}\mathbb{E}_{\mathcal{I}^*}\left[\sigma^2(X_i, A_i) + \{\hat{\mu}_i(\mathbf{O}_{i-1})(X_i, A_i) - \mu^*(X_i, A_i)\}^2\right]$$

$$= \sum_{i=1}^{n}\mathbb{E}_{\mathcal{I}^*}\left[\sigma^2(X_i, A_i)\right] + \sum_{i=1}^{n}\mathbb{E}_{\mathcal{I}^*}\left[\{\hat{\mu}_i(\mathbf{O}_{i-1})(X_i, A_i) - \mu^*(X_i, A_i)\}^2\right],$$

which leads to the following expression of the estimation error term (49):

$$\frac{1}{n}\sum_{i=1}^{n}\mathbb{E}_{\mathcal{I}^*}\left[\{\hat{\mu}_i(\mathbf{O}_{i-1})(X_i, A_i) - \mu^*(X_i, A_i)\}^2\right]$$

$$= \frac{1}{n}\mathbb{E}_{\mathcal{I}^*}\left[\sum_{i=1}^{n}\bar{l}_i\{\hat{\mu}_i(\mathbf{O}_{i-1})\}\right] - \frac{1}{n}\sum_{i=1}^{n}\mathbb{E}_{\mathcal{I}^*}\left[\sigma^2(X_i, A_i)\right] \tag{54}$$

$$= \frac{1}{n}\mathbb{E}_{\mathcal{I}^*}\left[\overline{\text{Regret}}(n; \overline{\mathcal{A}})\right] + \frac{1}{n}\mathbb{E}_{\mathcal{I}^*}\left[\inf\left\{\sum_{i=1}^{n}\bar{l}_i(\mu) : \mu \in \mathcal{F}\right\}\right] \tag{55}$$

$$- \frac{1}{n}\sum_{i=1}^{n}\mathbb{E}_{\mathcal{I}^*}\left[\sigma^2(X_i, A_i)\right].$$

Here, one may observe that

$$\frac{1}{n}\mathbb{E}_{\mathcal{I}^*}\left[\inf\left\{\sum_{i=1}^{n}\bar{l}_i(\mu) : \mu \in \mathcal{F}\right\}\right]$$

$$\leq \inf\left\{\frac{1}{n}\mathbb{E}_{\mathcal{I}^*}\left[\sum_{i=1}^{n}\bar{l}_i(\mu)\right] : \mu \in \mathcal{F}\right\}$$

$$= \inf\left\{\frac{1}{n}\sum_{i=1}^{n}\mathbb{E}_{\mathcal{I}^*}\left[\mathbb{E}_{\mathcal{I}^*}\left[\bar{l}_i(\mu)\middle|(\mathcal{F}_{i-1}, X_i, A_i)\right]\right] : \mu \in \mathcal{F}\right\} \tag{56}$$

$$\overset{\text{(a)}}{=} \inf\left\{\frac{1}{n}\sum_{i=1}^{n}\mathbb{E}_{\mathcal{I}^*}\left[\sigma^2(X_i, A_i) + \{\mu(X_i, A_i) - \mu^*(X_i, A_i)\}^2\right] : \mu \in \mathcal{F}\right\}$$

$$= \frac{1}{n}\sum_{i=1}^{n}\mathbb{E}_{\mathcal{I}^*}\left[\sigma^2(X_i, A_i)\right] + \inf\left\{\frac{1}{n}\sum_{i=1}^{n}\mathbb{E}_{\mathcal{I}^*}\left[\{\mu(X_i, A_i) - \mu^*(X_i, A_i)\}^2\right] : \mu \in \mathcal{F}\right\},$$

where the step (a) holds by the fact (51). Putting two pieces (54) and (56) together yields

$$
\frac{1}{n} \sum_{i=1}^{n} \mathbb{E}_{\mathcal{I}^*} \left[ \{ \hat{\mu}_i \left( \mathbf{O}_{i-1} \right) \left( X_i, A_i \right) - \mu^* \left( X_i, A_i \right) \}^2 \right]
$$

$$
\leq \frac{1}{n} \mathbb{E}_{\mathcal{I}^*} \left[ \overline{\text{Regret}} \left( n; \overline{\mathcal{A}} \right) \right] + \inf \left\{ \frac{1}{n} \sum_{i=1}^{n} \mathbb{E}_{\mathcal{I}^*} \left[ \{ \mu \left( X_i, A_i \right) - \mu^* \left( X_i, A_i \right) \}^2 \right] : \mu \in \mathcal{F} \right\}. \tag{57}
$$

Hence, the desired result (53) on the MSE of the AIPW estimator (8) is a straightforward consequence of the inequality (57) by plugging it into the bound (48). □

Here, we remark that aside from the optimal variance $v_*^2$, the MSE bound (53) shows two additional terms: (i) the expected regret relative to the number of rounds $n$, where the expectation is taken over $\mathbf{O}_n \sim \mathbb{P}_{\mathcal{I}^*}^n(\cdot)$; and (ii) the approximation error term whose form is slightly different from the one $\inf \left\{ \| \mu - \mu^* \|_{(n)}^2 : \mu \in \mathcal{F} \right\}$ appeared in the MSE bound (15) of Theorem 3.2.

**Non-asymptotic theory of online non-parametric regression**   Before delving into the investigation of the modified regret (52), we briefly recap the main results in [45] that establishes a theoretical framework of online non-parametric regression. In contrast to most existing works of online regression, the authors do NOT start from an algorithm, but instead directly work with the minimax regret in [45]. We will be able to extract a (not necessarily efficient) algorithm after taking a closer look at the minimax regret. Let us use $\langle\!\langle \cdots \rangle\!\rangle_{i=1}^n$ to denote an interleaved application of the operators inside repeated over $n$ rounds. With this notation in hand, the minimax regret of the online non-parametric regression problem for estimation of the treatment effect can be written as

$$
\mathcal{V}_n(\mathcal{F})
$$

$$
:= \left\langle\!\!\!\left\langle \sup_{(x_i, a_i) \in \mathbb{X} \times \mathbb{A}} \inf_{\hat{y}_i \in [-L, L]} \sup_{y_i \in [-L, L]} \right\rangle\!\!\!\right\rangle_{i=1}^n \left[ \sum_{i=1}^n (\hat{y}_i - y_i)^2 - \inf_{\mu \in \mathcal{F}} \sum_{i=1}^n \{ \mu(x_i, a_i) - y_i \}^2 \right], \tag{58}
$$

where $\mathcal{F} \subseteq (\mathbb{X} \times \mathbb{A} \to [-L, L])$ is a pre-specified function class. One of the key tools in the study of estimators based on i.i.d. data is the *symmetrization technique* [15, 62]. Under the i.i.d. scenario, one can investigate the supremum of an empirical process conditionally on the data by introducing Rademacher random variables, which is NOT directly applicable given the adaptive nature of our main problem. In the online prediction scenario, such a symmetrization technique becomes more subtle and it requires the notion of a binary tree, the smallest entity which captures the sequential nature of the problem in some sense. Towards achieving our goal in our problem, let us state some definitions.

**Definition B.1.**   Let $\mathbb{S}$ be a measurable state space. An $\mathbb{S}$-*valued tree of depth n* is a rooted complete binary tree with nodes labeled by elements of the state space $\mathbb{S}$: the sequence $\mathbf{s} = (\mathbf{s}_1, \mathbf{s}_2, \cdots, \mathbf{s}_n)$ of labeling functions $\mathbf{s}_i(\cdot) : \{\pm 1\}^{i-1} \to \mathbb{S}$ which provides the labels of each node. Here, $\mathbf{s}_1 \in \mathbb{S}$ is the label for the *root of the tree*, while $\mathbf{s}_i$ for $2 \leq i \leq n$ is the label of the node obtained by following the path of length $i-1$ from the root, with $+1$ indicating *right* and $-1$ indicating *left*. A *path of length n* is given by the sequence $\boldsymbol{\epsilon}_{1:n} = (\epsilon_1, \cdots, \epsilon_n) \in \{\pm 1\}^n$. Given any measurable function $\phi(\cdot) : \mathbb{S} \to \mathbb{R}$, $\phi(\mathbf{s})$ is an $\mathbb{R}$-valued tree of depth $n$ with labeling functions $(\phi \circ \mathbf{s}_i)(\cdot) : \{\pm 1\}^{i-1} \to \mathbb{R}$ for level $i \in [n]$ (or, in words, the evaluation of $\phi(\cdot) : \mathbb{S} \to \mathbb{R}$, $\phi(\mathbf{s})$ on $\mathbf{s}$). Lastly, we let Tree $(\mathbb{S}, n)$ denote the set of all $\mathbb{S}$-valued trees of depth $n$.

Here, one may think of the sequence of functions $\{\mathbf{s}_i(\cdot) : i \in [n]\}$ defined on the underlying sample space as a predictable stochastic process with respect to the dyadic filtration $\{\sigma(\boldsymbol{\epsilon}_{1:i}) : i \in [n]\}$. Next, let us define the notion of a *sequential $\beta$-cover* quantifies one of the key complexity measures of a function class $\mathcal{G} \subseteq (\mathbb{S} \to \mathbb{R})$ evaluated on the predictable process: the *sequential covering number*.

**Definition B.2** (Sequential covering numbers [46])**.**

(i) Define the following random pseudo-metric between two $\mathbb{R}$-valued trees $\mathbf{u} = (\mathbf{u}_i : i \in [n])$ and $\mathbf{v} = (\mathbf{v}_i : i \in [n])$ of depth $n$: for any $(p, \boldsymbol{\epsilon}_{1:n}) \in [1, +\infty] \times \{\pm 1\}^n$,

$$
d_{\boldsymbol{\epsilon}_{1:n}}^p(\mathbf{u}, \mathbf{v}) := \begin{cases} \left\{ \frac{1}{n} \sum_{i=1}^n |\mathbf{u}_i(\boldsymbol{\epsilon}_{1:i-1}) - \mathbf{v}_i(\boldsymbol{\epsilon}_{1:i-1})|^p \right\}^{\frac{1}{p}} & \text{if } 1 \leq p < +\infty; \\ \max \{|\mathbf{u}_i(\boldsymbol{\epsilon}_{1:i-1}) - \mathbf{v}_i(\boldsymbol{\epsilon}_{1:i-1})| : i \in [n]\} & \text{if } p = +\infty. \end{cases} \tag{59}
$$

(ii) A set $V \subseteq \text{Tree}(\mathbb{R}, n)$ is called a *sequential $\beta$-cover with respect to $l_p$-norm of $\mathcal{G} \subseteq (\mathbb{S} \to \mathbb{R})$ on a given $\mathbb{S}$-valued tree $\mathbf{s}$ of depth $n$*, where $p \in [1, +\infty]$, if

$$\sup \left\{ \inf \left\{ d^p_{\boldsymbol{\epsilon}_{1:n}}(\mathbf{u}, \mathbf{v}) : \mathbf{v} \in V \right\} : (\mathbf{u}, \boldsymbol{\epsilon}_{1:n}) \in \mathcal{G}(\mathbf{s}) \times \{\pm 1\}^n \right\} \leq \beta, \tag{60}$$

where $\mathcal{G}(\mathbf{s}) := \{g(\mathbf{s}) : g \in \mathcal{G}\} \subseteq \text{Tree}(\mathbb{R}, n)$;

(iii) The *sequential $\beta$-covering number with respect to $l_p$-norm of a function class $\mathcal{G} \subseteq (\mathbb{S} \to \mathbb{R})$ on an $\mathbb{S}$-valued tree $\mathbf{s}$ of depth $n$*, where $p \in [1, +\infty]$, is defined by

$$\begin{aligned}&\mathcal{N}_p(\beta, \mathcal{G}, \mathbf{s})\\&:= \min\left\{|V| : V \subseteq \text{Tree}(\mathbb{R}, n) \text{ is a sequential } \beta\text{-cover w.r.t. } l_p\text{-norm of } \mathcal{G} \text{ on } \mathbf{s}\right\}.\end{aligned}$$

Let us further define $\mathcal{N}_p(\beta, \mathcal{G}, n) := \sup\{\mathcal{N}_p(\beta, \mathcal{G}, \mathbf{s}) : \mathbf{s} \in \text{Tree}(\mathbb{S}, n)\}$ to be the *maximal sequential $\beta$-covering number with respect to $l_p$-norm of $\mathcal{G}$ over $\mathbb{S}$-valued trees of depth $n$*. Now, we will refer to $\log \mathcal{N}_p(\beta, \mathcal{G}, n)$ as the *sequential $\beta$-metric entropy of $\mathcal{G}$ with respect to $l_p$-norm*.

In particular, we are going to study the behavior of the minimax regret $\mathcal{V}_n(\mathcal{F})$ for the case where the sequential metric entropy of $\mathcal{F} \subseteq (\mathbb{X} \times \mathbb{A} \to [-L, L])$ w.r.t. $l_2$-norm grows polynomially as the scale $\beta$ decreases:

$$\log \mathcal{N}_2(\beta, \mathcal{F}, n) \sim \beta^{-p} \quad \text{for } p \in (0, +\infty). \tag{61}$$

Let us also consider the *parametric "$p = 0$" case* when the sequential covering number of $\mathcal{F}$ with respect to $l_2$-norm itself behaves as:

$$\mathcal{N}_2(\beta, \mathcal{F}, n) \sim \beta^{-d}. \tag{62}$$

For instance, the function class $\mathcal{F} := \{f_{\boldsymbol{\theta}}(\cdot) : \mathbb{R}^d \to \mathbb{R} : \boldsymbol{\theta} \in \Theta\}$ for the linear regression problem in a bounded measurable subset $\Theta \subseteq \mathbb{R}^d$, where the function $f_{\boldsymbol{\theta}}(\cdot) : \mathbb{R}^d \to \mathbb{R}$ is given by $f_{\boldsymbol{\theta}}(\mathbf{x}) := \boldsymbol{\theta}^\top \mathbf{x}$ for $\boldsymbol{\theta} \in \mathbb{R}^d$, satisfies the condition (62). By employing the main results (in particular, *Theorem 2*) in [45], one can establish the following conclusion:

**Theorem B.2** (The rates of convergence of the minimax regret). *Given any function class $\mathcal{F} \subseteq (\mathbb{X} \times \mathbb{A} \to [-L, L])$ with sequential metric entropy growth $\log \mathcal{N}_2(\beta, \mathcal{F}, n) \leq \beta^{-p}$ for $p \in (0, +\infty)$, it holds that*

*(i) for $p \in (2, +\infty)$, the minimax regret (58) is bounded as*

$$\mathcal{V}_n(\mathcal{F}) \leq \left(4 + \frac{24}{p-2}\right) Ln^{1-\frac{1}{p}}. \tag{63}$$

*(ii) for $p \in (0, 2)$, the minimax regret (58) is bounded as*

$$\mathcal{V}_n(\mathcal{F}) \leq \left(32L^2 + 4L + \frac{24L}{2-p}\right) n^{1-\frac{2}{p+2}}. \tag{64}$$

*(iii) for $p = 2$, the minimax regret (58) is bounded as*

$$\mathcal{V}_n(\mathcal{F}) \leq \left(32L^2 + 4L + 3\right) \sqrt{n} \log n. \tag{65}$$

*(iv) for the parametric case (62), the minimax regret (58) is bounded as*

$$\mathcal{V}_n(\mathcal{F}) \leq \left(16L^2 + 4L + 12\right) d \log n. \tag{66}$$

*(v) if the function class $\mathcal{F} \subseteq (\mathbb{X} \times \mathbb{A} \to [-L, L])$ is a finite set, the minimax regret (58) is bounded as*

$$\mathcal{V}_n(\mathcal{F}) \leq 32L^2 \log |\mathcal{F}|. \tag{67}$$

It is shown in [45] that the upper bounds (i)–(iv) on the minimax regret (58) in Theorem B.2 are *tight up to logarithmic factors*. See *Theorem 3* therein for further details.

Although Theorem B.2 characterizes the rates of convergence of the minimax regret (58) in various scenarios *statistically*, its proof is *non-constructive* in the sense that the regret bounds therein are established without explicitly constructing an algorithm. In order to provide a general algorithmic framework for the problem of online non-parametric regression, we follow the abstract *relaxation recipe* proposed in [47]. It was shown in [47] that if one can find a sequence of mappings from the observed data to real numbers $\mathrm{Rel}_n$, often called a *relaxation*, satisfying some desirable conditions, then one can construct estimators based on such relaxations. To be specific, we search for a relaxation $\mathrm{Rel}_n\left(\cdot,\cdot\right):\biguplus_{k=0}^n\left\{\left(\mathbb{X}\times\mathbb{A}\right)^k\times\left[-L,L\right]^k\right\}\to\mathbb{R}$ that satisfies the following two conditions:

**Assumption 5** (Initial condition). The relaxation $\mathrm{Rel}_n\left(\cdot,\cdot\right):\biguplus_{k=0}^n\left\{\left(\mathbb{X}\times\mathbb{A}\right)^k\times\left[-L,L\right]^k\right\}\to\mathbb{R}$ satisfies

$$\mathrm{Rel}_n\left(\left(\mathbf{x},\mathbf{a}\right)_{1:n},\mathbf{y}_{1:n}\right)\geq-\inf\left\{\sum_{k=1}^n\left\{y_i-\mu\left(x_i,a_i\right)\right\}^2:\mu(\cdot,\cdot)\in\mathcal{F}\right\},\tag{68}$$

where $\left(\mathbf{x},\mathbf{a}\right)_{1:k}:=\left(\left(x_i,a_i\right):i\in[k]\right)\in\left(\mathbb{X}\times\mathbb{A}\right)^k$ and $\mathbf{y}_{1:k}:=\left(y_i:i\in[k]\right)\in\left[-L,L\right]^k$ for every $k\in[n]$.

**Assumption 6** (Recursive admissibility condition). The relaxation $\mathrm{Rel}_n\left(\cdot,\cdot\right)$ satisfies

$$\inf_{\hat{y}_k\in[-L,L]}\sup_{y_k\in[-L,L]}\left\{\left(\hat{y}_k-y_k\right)^2+\mathrm{Rel}_n\left(\left(\mathbf{x},\mathbf{a}\right)_{1:k},\mathbf{y}_{1:k}\right)\right\}\leq\mathrm{Rel}_n\left(\left(\mathbf{x},\mathbf{a}\right)_{1:k-1},\mathbf{y}_{1:k-1}\right),\tag{69}$$

for any $k\in[n]$ and any $x_k\in\mathbb{X}$.

A relaxation $\mathrm{Rel}_n\left(\cdot,\cdot\right):\biguplus_{k=0}^n\left\{\left(\mathbb{X}\times\mathbb{A}\right)^k\times\left[-L,L\right]^k\right\}\to\mathbb{R}$ satisfying Assumptions 5 and 6 is said to be *admissible*. With an admissible relaxation $\mathrm{Rel}_n\left(\cdot,\cdot\right)$ in hand, one can design an algorithm for the online non-parametric regression problem with the following associated regret bound (see Algorithm 5 for a detailed description):

$$\overline{\mathrm{Regret}}\left(n,\mathcal{F};\mathrm{Alg.}\ 5\right)$$

$$=\sum_{i=1}^n\left\{Y_i-\hat{\mu}_i\left(\mathbf{O}_{i-1}\right)\left(X_i,A_i\right)\right\}^2-\inf\left\{\sum_{i=1}^n\left\{Y_i-\mu\left(X_i,A_i\right)\right\}^2:\mu\in\mathcal{F}\right\}\tag{70}$$

$$\leq\mathrm{Rel}_n\left(\varnothing,\varnothing\right).$$

We further notice that if the function $y_i\in[-L,L]\mapsto\left(\hat{y}-y_i\right)^2+\mathrm{Rel}_n\left(\left(\left(\mathbf{x},\mathbf{a}\right)_{1:i}\right),\left(\mathbf{y}_{1:i-1},y_i\right)\right)$ is convex for every $\left(\hat{y},\mathbf{x}_{1:n},\mathbf{a}_{1:n},\mathbf{y}_{1:i-1}\right)\in[-L,L]\times\mathbb{X}^n\times\mathbb{A}^n\times\left[-L,L\right]^{i-1}$ and $i\in[n]$, then the prediction rules (71) and (72) becomes much simpler, since the supremum over $y_i\in[-L,L]$ is attained either $L$ or $-L$. The prediction rules then can be written as

$$\hat{\mu}_1(\varnothing)(x,a)$$
$$\in\arg\min\left\{\max\left\{\left(\hat{y}-L\right)^2+\mathrm{Rel}_n\left(\left(x,a\right),L\right),\left(\hat{y}+L\right)^2+\mathrm{Rel}_n\left(\left(x,a\right),-L\right)\right\}:\hat{y}\in[-L,L]\right\},\tag{73}$$

and for $i\in\{2,3,\cdots,n\}$,

$$\hat{\mu}_i\left(\mathbf{O}_{i-1}\right)\left(x,a\right)$$
$$\in\arg\min\left\{\max\left\{\left(\hat{y}-L\right)^2+\mathrm{Rel}_n\left(\left(\left(\mathbf{X},\mathbf{A}\right)_{1:i-1},\left(x,a\right)\right),\left(\mathbf{Y}_{1:i-1},L\right)\right),\right.\right.\tag{74}$$
$$\left.\left.\left(\hat{y}+L\right)^2+\mathrm{Rel}_n\left(\left(\left(\mathbf{X},\mathbf{A}\right)_{1:i-1},\left(x,a\right)\right),\left(\mathbf{Y}_{1:i-1},-L\right)\right)\right\}:\hat{y}\in[-L,L]\right\}.$$

One can easily observe that the prediction rules (73) and (74) can be further simplified as

$$\hat{\mu}_1(\varnothing)(x,a)=\chi_{[-L,L]}\left\{\frac{\mathrm{Rel}_n\left(\left(x,a\right),L\right)-\mathrm{Rel}_n\left(\left(x,a\right),-L\right)}{4L}\right\},\tag{75}$$

**Algorithm 5** A generic forecaster based on the relaxation recipe proposed in [47]

**Require:** a relaxation $\mathrm{Rel}_n\left(\cdot,\cdot\right): \biguplus_{k=0}^{n}\left\{\left(\mathbb{X}\times\mathbb{A}\right)^{k}\times\left[-L,L\right]^{k}\right\}\rightarrow\mathbb{R}$.

1: We first choose $\hat{\mu}_1(\varnothing)(\cdot,\cdot)\in(\mathbb{X}\times\mathbb{A}\rightarrow\mathbb{R})$ as

$$\hat{\mu}_1(\varnothing)(x,a)\in\left\{\sup_{y_1\in[-L,L]}\left\{\left(\hat{y}-y_1\right)^2+\mathrm{Rel}_n\left(\left(x,a\right),y_1\right)\right\}:\hat{y}\in[-L,L]\right\}.\qquad(71)$$

2: **for** $i=2,3,\cdots,n$, **do**
3:     Observe a triple $(X_i,A_i,Y_i)\in\mathbb{O}$;
4:     We compute $\hat{\mu}_i\left(\mathbf{O}_{i-1}\right)\in(\mathbb{X}\times\mathbb{A}\rightarrow\mathbb{R})$ according to the following rule:

$$\hat{\mu}_i\left(\mathbf{O}_{i-1}\right)(x,a)$$
$$\in\arg\min\left\{\sup_{y_i\in[-L,L]}\left\{\left(\hat{y}-y_i\right)^2+\mathrm{Rel}_n\left(\left(\left(\mathbf{X},\mathbf{A}\right)_{1:i-1},(x,a)\right),\left(\mathbf{Y}_{1:i-1},y_i\right)\right)\right\}:\hat{y}\in[-L,L]\right\}.$$
$$(72)$$

5: **end for**
6: **return** the sequence of estimates $\{\hat{\mu}_i\left(\mathbf{O}_{i-1}\right)\in(\mathbb{X}\times\mathbb{A}\rightarrow\mathbb{R}):i\in[n]\}$ of the treatment effect.

and for $i\in\{2,3,\cdots,n\}$,

$$\hat{\mu}_i\left(\mathbf{O}_{i-1}\right)(x,a)$$
$$=\chi_{[-L,L]}\left\{\frac{\mathrm{Rel}_n\left(\left(\left(\mathbf{X},\mathbf{A}\right)_{1:i-1},(x,a)\right),\left(\mathbf{Y}_{1:i-1},L\right)\right)-\mathrm{Rel}_n\left(\left(\left(\mathbf{X},\mathbf{A}\right)_{1:i-1},(x,a)\right),\left(\mathbf{Y}_{1:i-1},-L\right)\right)}{4L}\right\},$$
$$(76)$$

where $\chi_{[-L,L]}(\cdot):\mathbb{R}\rightarrow[-L,L]$ defines a clip function onto the interval $[-L,L]$, i.e.,

$$\chi_{[-L,L]}(x):=\begin{cases}L & \text{if } x>L;\\ x & \text{if } -L\le x\le L;\\ -L & \text{otherwise.}\end{cases}$$

By directly using *Lemma 16* in [45], one can obtain the following significant result:

**Theorem B.3.** *The relaxation* $\mathcal{R}_n\left(\cdot,\cdot\right):\biguplus_{k=0}^{n}\left\{\left(\mathbb{X}\times\mathbb{A}\right)^{k}\times\left[-L,L\right]^{k}\right\}\rightarrow\mathbb{R}$ *defined as*

$$\mathcal{R}_n\left(\left(\mathbf{x},\mathbf{a}\right)_{1:k},\mathbf{y}_{1:k}\right)$$
$$:=\sup_{(\mathbf{z},\mathbf{m})}\mathbb{E}_{\boldsymbol{\epsilon}_{1:n}\sim\mathrm{Unif}(\{\pm1\}^n)}\left[\sup\left\{\sum_{j=k+1}^{n}\left[4L\epsilon_j\left\{\mu\left(\mathbf{z}_j\left(\boldsymbol{\epsilon}_{1:j-1}\right)\right)-\mathbf{m}_j\left(\boldsymbol{\epsilon}_{1:j-1}\right)\right\}\right.\right.\right.$$
$$\left.\left.\left.-\left\{\mu\left(\mathbf{z}_j\left(\boldsymbol{\epsilon}_{1:j-1}\right)\right)-\mathbf{m}_j\left(\boldsymbol{\epsilon}_{1:j-1}\right)\right\}^2\right]-\sum_{j=1}^{k}\left\{\mu\left(x_j,a_j\right)-y_j\right\}^2:\mu\in\mathcal{F}\right\}\right],$$
$$(77)$$

*where the pair* $(\mathbf{z},\mathbf{m})$ *ranges over the set* $\mathrm{Tree}\left(\mathbb{X}\times\mathbb{A},n\right)\times\mathrm{Tree}\left(\mathbb{R},n\right)$, *is an admissible relaxation. As a direct consequence of the regret bound* (70), *Algorithm 5 using the admissible relaxation* $\mathcal{R}_n\left(\cdot,\cdot\right)$ *as an input enjoys the regret bound of an offset Rademacher complexity:*

$$\overline{\mathrm{Regret}}\left(n,\mathcal{F};\text{Alg. }5\right)$$
$$\le\mathcal{R}_n\left(\varnothing,\varnothing\right)$$
$$=\sup_{(\mathbf{z},\mathbf{m})}\mathbb{E}_{\boldsymbol{\epsilon}_{1:n}\sim\mathrm{Unif}(\{\pm1\}^n)}\left[\sup\left\{\sum_{j=1}^{n}\left[4L\epsilon_j\left\{\mu\left(\mathbf{z}_j\left(\boldsymbol{\epsilon}_{1:j-1}\right)\right)-\mathbf{m}_j\left(\boldsymbol{\epsilon}_{1:j-1}\right)\right\}\right.\right.\right.$$
$$\left.\left.\left.-\left\{\mu\left(\mathbf{z}_j\left(\boldsymbol{\epsilon}_{1:j-1}\right)\right)-\mathbf{m}_j\left(\boldsymbol{\epsilon}_{1:j-1}\right)\right\}^2\right]:\mu\in\mathcal{F}\right\}\right].$$
$$(78)$$

Since the upper bounds on the minimax regret (58) provided in Theorem B.2 are established by further upper bounding the offset Rademacher complexity $\mathcal{R}_n(\varnothing, \varnothing)$, one can end up with the following corollary:

**Corollary B.1.** *Consider any function class $\mathcal{F} \subseteq (\mathbb{X} \times \mathbb{A} \to [-L, L])$ with sequential metric entropy growth $\log \mathcal{N}_2(\beta, \mathcal{F}, n) \leq \beta^{-p}$ for $p \in (0, +\infty)$. Then, Algorithm 5 using the admissible relaxation $\mathcal{R}_n(\cdot, \cdot)$ defined by (77) as an input enjoys the following regret bounds:*

*(i) for $p \in (2, +\infty)$, it holds that*

$$\overline{\mathrm{Regret}}(n, \mathcal{F}; \mathrm{Alg. 5}) \leq \left( 4 + \frac{24}{p-2} \right) L n^{1 - \frac{1}{p}}. \tag{79}$$

*(ii) for $p \in (0, 2)$, it holds that*

$$\overline{\mathrm{Regret}}(n, \mathcal{F}; \mathrm{Alg. 5}) \leq \left( 32L^2 + 4L + \frac{24L}{2-p} \right) n^{1 - \frac{2}{p+2}}. \tag{80}$$

*(iii) for $p = 2$, it holds that*

$$\overline{\mathrm{Regret}}(n, \mathcal{F}; \mathrm{Alg. 5}) \leq \left( 32L^2 + 4L + 3 \right) \sqrt{n} \log n. \tag{81}$$

*(iv) for the parametric case (62), it holds that*

$$\overline{\mathrm{Regret}}(n, \mathcal{F}; \mathrm{Alg. 5}) \leq \left( 16L^2 + 4L + 12 \right) d \log n. \tag{82}$$

*(v) if the function class $\mathcal{F} \subseteq (\mathbb{X} \times \mathbb{A} \to [-L, L])$ is a finite set, it holds that*

$$\overline{\mathrm{Regret}}(n, \mathcal{F}; \mathrm{Alg. 5}) \leq 32L^2 \log |\mathcal{F}|. \tag{83}$$

Even though Corollary B.1 gives no-regret learning guarantees of Algorithm 5 with the admissible relaxation $\mathcal{R}_n(\cdot, \cdot)$ defined by (77) for various function classes $\mathcal{F} \subseteq (\mathbb{X} \times \mathbb{A} \to [-L, L])$, it is still NOT a practical algorithm since the relaxation $\mathcal{R}_n(\cdot, \cdot)$ defined as (77) is not directly computable in general. To address this problem, [45] provided a generic schema for deriving implementable online non-parametric regression algorithms. The schema can be described as follows:

(a) Find a *computable relaxation* $\mathrm{Rel}_n(\cdot, \cdot) : \biguplus_{k=0}^n \left\{ (\mathbb{X} \times \mathbb{A})^k \times [-L, L]^k \right\} \to \mathbb{R}$ such that

$$\mathcal{R}_n((\mathbf{x}, \mathbf{a})_{1:k}, \mathbf{y}_{1:k}) \leq \mathrm{Rel}_n((\mathbf{x}, \mathbf{a})_{1:k}, \mathbf{y}_{1:k})$$

for every $(k, \mathbf{x}_{1:n}, \mathbf{a}_{1:n}, \mathbf{y}_{1:n}) \in \{0, 1, \cdots, n\} \times \mathbb{X}^n \times \mathbb{A}^n \times [-L, L]^n$, and the function $y_k \in [-L, L] \mapsto (\hat{y} - y_k)^2 + \mathrm{Rel}_n(((\mathbf{x}, \mathbf{a})_{1:k}), (\mathbf{y}_{1:k-1}, y_k)) \in \mathbb{R}$ is convex for every $(\hat{y}, \mathbf{x}_{1:n}, \mathbf{a}_{1:n}, \mathbf{y}_{1:k-1}) \in [-L, L] \times \mathbb{X}^n \times \mathbb{A}^n \times [-L, L]^{k-1}$ and $k \in [n]$;

(b) Next, we check the following condition:

$$\sup_{(x_k, a_k, \mu_k) \in \mathbb{X} \times \mathbb{A} \times \Delta([-L,L])} \left\{ \mathbb{E}_{y_k \sim \mu_k} \left[ \left( \mathbb{E}_{y_k \sim \mu_k} [y_k] - y_k \right)^2 \right] + \mathbb{E}_{y_k \sim \mu_k} \left[ \mathrm{Rel}_n((\mathbf{x}, \mathbf{a})_{1:k}, \mathbf{y}_{1:k}) \right] \right\}$$

$$\leq \mathrm{Rel}_n((\mathbf{x}, \mathbf{a})_{1:k-1}, \mathbf{y}_{1:k-1})$$

for every $(\mathbf{x}_{1:k-1}, \mathbf{a}_{1:k-1}, \mathbf{y}_{1:k-1}) \in \mathbb{X}^{k-1} \times \mathbb{A}_{k-1} \times [-L, L]^{k-1}$ and $k \in [n]$;

(c) Implement Algorithm 5 using the relaxation $\mathrm{Rel}_n(\cdot, \cdot)$ as an input.

The authors proved that any computable relaxation $\mathrm{Rel}_n(\cdot, \cdot)$ satisfying conditions stated in (a) and (b) are admissible; see *Proposition 17* therein. Consequently, any online non-parametric regression algorithm produced by the above generic schema always satisfies the regret bound (70). Moreover, the authors established a practical online non-parametric regression algorithm with no-regret learning guarantees based on the above schema for the finite function class $\mathcal{F} \subseteq (\mathbb{X} \times \mathbb{A} \to [-L, L])$ and the online linear regression problem.

# C Proofs for Section 4

## C.1 Proof of Theorem 4.1

Theorem 4.1 can be established by taking the following two lemmas collectively:

**Lemma C.1.** *Under Assumption 3, the local minimax risk over the class $\mathcal{C}_\delta\left(\mathcal{I}^*\right)$ is lower bounded by*

$$\mathcal{M}_n\left(\mathcal{C}_\delta\left(\mathcal{I}^*\right)\right) \geq \frac{1}{2304}\left(1 - \frac{1}{\sqrt{2}}\right) \cdot \frac{1}{n}\mathrm{Var}_{X \sim \Xi^*}\left[\langle g(X, \cdot), \mu^*(X, \cdot)\rangle_{\lambda_{\mathbb{A}}}\right], \tag{84}$$

*provided that $n \geq 16H_{2\to4}^2$.*

**Lemma C.2.** *Under Assumption 4, the local minimax risk over the class $\mathcal{C}_\delta\left(\mathcal{I}^*\right)$ is lower bounded by*

$$\mathcal{M}_n\left(\mathcal{C}_\delta\left(\mathcal{I}^*\right)\right) \geq \frac{1}{8K^4} \cdot \frac{\|\sigma\|_{(n)}^2}{n}. \tag{85}$$

## C.2 Proof of Lemma C.1

The proof relies on Le Cam's two-point method by taking the outcome kernel $\Gamma^* : \mathbb{X} \times \mathbb{A} \to \Delta(\mathbb{Y})$ to be fixed, and perturbing the context distribution $\Xi^*(\cdot) \in \Delta(\mathbb{X})$: we first construct a collection of context distributions $\{\Xi_s(\cdot) \in \Delta(\mathbb{X}) : s \in (0, +\infty)\}$. Later, we will choose the parameter $s > 0$ small enough so that $\Xi_s \in \mathcal{N}\left(\Xi^*\right)$ and two distributions $\mathbb{P}_{(\Xi_s,\Gamma^*)}^n \in \Delta\left(\mathbb{O}^n\right)$ and $\mathbb{P}_{(\Xi^*,\Gamma^*)}^n \in \Delta\left(\mathbb{O}^n\right)$ are *indistinguishable*, but large enough such that the functional values $\tau\left(\Xi_s, \Gamma^*\right)$ and $\tau\left(\Xi^*, \Gamma^*\right)$ are *well-separated*. Le Cam's two-point lemma (the equation (15.14) in [62]) guarantees that the local minimax risk $\mathcal{M}_n\left(\mathcal{C}_\delta\left(\mathcal{I}^*\right)\right)$ is lower bounded as

$$\mathcal{M}_n\left(\mathcal{C}_\delta\left(\mathcal{I}^*\right)\right) \geq \frac{1}{4}\left\{1 - \mathrm{TV}\left(\mathbb{P}_{(\Xi_s,\Gamma^*)}^n, \mathbb{P}_{(\Xi^*,\Gamma^*)}^n\right)\right\}\left\{\tau\left(\Xi_s, \Gamma^*\right) - \tau\left(\Xi^*, \Gamma^*\right)\right\}^2, \tag{86}$$

provided that $\Xi_s \in \mathcal{N}\left(\Xi^*\right)$.

As the first step, we upper bound the total variation distance $\mathrm{TV}\left(\mathbb{P}_{(\Xi_s,\Gamma^*)}^n, \mathbb{P}_{(\Xi^*,\Gamma^*)}^n\right)$. Thanks to the Pinsker-Csiszár-Kullback inequality, one has

$$\mathrm{TV}\left(\mathbb{P}_{(\Xi_s,\Gamma^*)}^n, \mathbb{P}_{(\Xi^*,\Gamma^*)}^n\right) \leq \sqrt{\frac{1}{2}\mathrm{KL}\left(\mathbb{P}_{(\Xi_s,\Gamma^*)}^n \middle\| \mathbb{P}_{(\Xi^*,\Gamma^*)}^n\right)}. \tag{87}$$

We can find that the density function of the law $\mathbb{P}_{\mathcal{I}}^n = \mathbb{P}_{(\Xi,\Gamma)}^n \in \Delta\left(\mathbb{O}^n\right)$ of the sample trajectory $\mathbf{O}_n$ under the problem instance $\mathcal{I} = (\Xi, \Gamma) \in \mathbb{I}$ with respect to the base measure $\left(\lambda_{\mathbb{X}} \otimes \lambda_{\mathbb{A}} \otimes \lambda_{\mathbb{A}}\right)^{\otimes n}$ is given by

$$p_{\mathcal{I}}^n\left(\mathbf{o}_n\right) = p_{(\Xi,\Gamma)}^n\left(\mathbf{o}_n\right) = \prod_{i=1}^n\left\{\xi\left(x_i\right)\pi_i^*\left(x_i, \mathbf{o}_{i-1}; a_i\right)\gamma\left(y_i\middle| x_i, a_i\right)\right\}. \tag{88}$$

Using this fact, the KL-divergence $\mathrm{KL}\left(\mathbb{P}_{(\Xi_s,\Gamma^*)}^n \middle\| \mathbb{P}_{(\Xi^*,\Gamma^*)}^n\right)$ can be computed as

$$\begin{aligned}
&\mathrm{KL}\left(\mathbb{P}_{(\Xi_s,\Gamma^*)}^n \middle\| \mathbb{P}_{(\Xi^*,\Gamma^*)}^n\right) \\
&= \mathbb{E}_{(\Xi_s,\Gamma^*)}\left[\log\frac{p_{(\Xi_s,\Gamma^*)}^n\left(\mathbf{O}_n\right)}{p_{(\Xi^*,\Gamma^*)}^n\left(\mathbf{O}_n\right)}\right] \\
&= \mathbb{E}_{(\Xi_s,\Gamma^*)}\left[\sum_{i=1}^n\log\frac{\xi_s\left(X_i\right)\pi_i^*\left(X_i, \mathbf{O}_{i-1}; A_i\right)\gamma^*\left(Y_i\middle| X_i, A_i\right)}{\xi^*\left(X_i\right)\pi_i^*\left(X_i, \mathbf{O}_{i-1}; A_i\right)\gamma^*\left(Y_i\middle| X_i, A_i\right)}\right] \\
&= \sum_{i=1}^n\mathbb{E}_{(\Xi_s,\Gamma^*)}\left[\log\frac{\xi_s\left(X_i\right)}{\xi^*\left(X_i\right)}\right] \\
&= n \cdot \mathrm{KL}\left(\Xi_s\middle\|\Xi^*\right).
\end{aligned} \tag{89}$$

So if one can show that $\Xi_s \in \mathcal{N}(\Xi^*)$, then the equation (89) guarantees that

$$\mathrm{KL}\left(\mathbb{P}^n_{(\Xi_s, \Gamma^*)} \middle\| \mathbb{P}^n_{(\Xi^*, \Gamma^*)}\right) = n \cdot \mathrm{KL}\left(\Xi_s \middle\| \Xi^*\right) \leq 1,$$

which can be taken collectively with the bound (87) to produce the following conclusion:

$$\mathrm{TV}\left(\mathbb{P}^n_{(\Xi_s, \Gamma^*)}, \mathbb{P}^n_{(\Xi^*, \Gamma^*)}\right) \leq \frac{1}{\sqrt{2}}. \tag{90}$$

With the arguments thus far in place, it remains to construct a family $\{\Xi_s \in \Delta(\mathbb{X}) : s \in (0, +\infty)\}$ and then choose a parameter $s > 0$ such that $\Xi_s \in \mathcal{N}(\Xi^*)$ and the functional values $\tau(\Xi_s, \Gamma^*)$ and $\tau(\Xi^*, \Gamma^*)$ are well-separated. To this end, we consider the function $\tilde{h}(\cdot) : \mathbb{X} \to \mathbb{R}$ defined by

$$\tilde{h}(x) := \begin{cases} h(x) & \text{if } |h(x)| \leq 2H_{2\to 4}\sqrt{\mathbb{E}_{X \sim \Xi^*}[h^2(X)]}; \\ \mathrm{sign}(h(x))\sqrt{\mathbb{E}_{X \sim \Xi^*}[h^2(X)]} & \text{otherwise.} \end{cases}$$

Since $H_{2\to 4} \geq 1$, one can easily find that $\left|\tilde{h}(x)\right| \leq |h(x)|$ for all $x \in \mathbb{X}$. Now for each $s \in (0, +\infty)$, we define the *tilted probability measure* $\Xi_s(\cdot) \in \Delta(\mathbb{X})$ by

$$\xi_s(x) = \frac{\mathrm{d}\Xi_s}{\mathrm{d}\lambda_{\mathbb{X}}}(x) := \frac{1}{\mathcal{Z}(s)}\xi^*(x)\exp\left(s\tilde{h}(x)\right), \quad \forall x \in \mathbb{X}, \tag{91}$$

where $\mathcal{Z}(s) := \int_{\mathbb{X}} \xi^*(x)\exp\left(s\tilde{h}(x)\right)\mathrm{d}\lambda_{\mathbb{X}}(x) = \mathbb{E}_{X \sim \Xi^*}\left[\exp\left(s\tilde{h}(X)\right)\right]$. At this point, we note for every $x \in \mathbb{X}$ that

$$\exp\left(-s\left\|\tilde{h}\right\|_\infty\right) \leq \exp\left(s\tilde{h}(x)\right) \leq \exp\left(s\left\|\tilde{h}\right\|_\infty\right), \tag{92}$$

which also immediately yields

$$\exp\left(-s\left\|\tilde{h}\right\|_\infty\right) \leq \mathcal{Z}(s) = \mathbb{E}_{X \sim \Xi^*}\left[\exp\left(s\tilde{h}(X)\right)\right] \leq \exp\left(s\left\|\tilde{h}\right\|_\infty\right). \tag{93}$$

Here, we choose $s = \frac{1}{4\|h\|_{L^2(\Xi^*)}\sqrt{n}} > 0$. Then, it holds due to the fact $\left|\tilde{h}(x)\right| \leq 2H_{2\to 4}\|h\|_{L^2(\Xi^*)}$ for all $x \in \mathbb{X}$ that

$$s\left\|\tilde{h}\right\|_\infty = \frac{1}{4\sqrt{n}} \cdot \frac{\left\|\tilde{h}\right\|_\infty}{\|h\|_{L^2(\Xi^*)}} \leq \frac{H_{2\to 4}}{2\sqrt{n}} \overset{\text{(a)}}{\leq} \frac{1}{8}, \tag{94}$$

where the step (a) follows due to the assumption that $n \geq 16H^2_{2\to 4}$. Now, it's time to prove that $\Xi_s \in \mathcal{N}(\Xi^*)$ for the current choice of the parameter $s > 0$. Due to Theorem 5 in [14], it follows that

$$\mathrm{KL}\left(\Xi_s \middle\| \Xi^*\right) \leq \log\left\{1 + \chi^2\left(\Xi_s \middle\| \Xi^*\right)\right\} \leq \chi^2\left(\Xi_s \middle\| \Xi^*\right). \tag{95}$$

So it suffices to upper bound the $\chi^2$-divergence $\chi^2\left(\Xi_s \middle\| \Xi^*\right)$. One can reveal that

$$\begin{aligned}
\chi^2\left(\Xi_s \middle\| \Xi^*\right) &= \mathrm{Var}_{X \sim \Xi^*}\left[\frac{\xi_s(X)}{\xi^*(X)}\right] \\
&= \frac{1}{\mathcal{Z}^2(s)}\mathrm{Var}_{X \sim \Xi^*}\left[\exp\left(s\tilde{h}(X)\right)\right] \\
&\leq \frac{1}{\mathcal{Z}^2(s)}\mathbb{E}_{X \sim \Xi^*}\left[\left\{\exp\left(s\tilde{h}(X)\right) - 1\right\}^2\right] \\
&\overset{\text{(b)}}{\leq} \exp\left(2s\left\|\tilde{h}\right\|_\infty\right)\mathbb{E}_{X \sim \Xi^*}\left[\exp\left(2s\left|\tilde{h}(X)\right|\right) \cdot s^2\tilde{h}^2(X)\right] \\
&\overset{\text{(c)}}{\leq} \exp\left(4s\left\|\tilde{h}\right\|_\infty\right) \cdot s^2\mathbb{E}_{X \sim \Xi^*}\left[h^2(X)\right],
\end{aligned} \tag{96}$$

where the step (b) makes use of the fact (93) together with the elementary bound $|\exp(u) - 1| \leq |u|\exp(|u|), \forall u \in \mathbb{R}$, and the step (c) follows from the fact $\left|\tilde{h}(x)\right| \leq |h(x)|, \forall x \in \mathbb{X}$. If we put

$s = \frac{1}{4\|h\|_{L^2(\Xi^*)}\sqrt{n}}$ into the bound (96), then we obtain from the fact $s\left\|\tilde{h}\right\|_\infty \le \frac{1}{8}$ together with the basic inequality (95) that

$$\mathrm{KL}\left(\Xi_s\,\|\,\Xi^*\right) \le \chi^2\left(\Xi_s\,\|\,\Xi^*\right) \le 2s^2\,\|h\|^2_{L^2(\Xi^*)} = \frac{1}{8n}, \tag{97}$$

which implies $\Xi_s \in \mathcal{N}\left(\Xi^*\right)$ for the choice of the parameter $s = \frac{1}{4\|h\|_{L^2(\Xi^*)}\sqrt{n}}$. Hence, the upper bound on the total variation distance (90) turns out to be valid.

Next, we lower bound the gap between the functional values $\tau\left(\Xi_s, \Gamma^*\right)$ and $\tau\left(\Xi^*, \Gamma^*\right)$. It holds that

$$
\begin{aligned}
&\tau\left(\Xi_s, \Gamma^*\right) - \tau\left(\Xi^*, \Gamma^*\right) \\
&= \mathbb{E}_{X\sim\Xi_s}\left[\langle g(X,\cdot), \mu^*(X,\cdot)\rangle_{\lambda_\mathbb{A}}\right] - \tau\left(\mathcal{I}^*\right) \\
&= \frac{1}{\mathcal{Z}(s)}\int_\mathbb{X} \xi^*(x)\exp\left(s\tilde{h}(x)\right)\underbrace{\left\{\langle g(x,\cdot), \mu^*(x,\cdot)\rangle_{\lambda_\mathbb{A}} - \tau\left(\mathcal{I}^*\right)\right\}}_{=\,h(x)}\,\mathrm{d}\lambda_\mathbb{X}(x) \\
&= \frac{1}{\mathcal{Z}(s)}\mathbb{E}_{X\sim\Xi^*}\left[h(X)\exp\left(s\tilde{h}(X)\right)\right] \\
&= \frac{\mathbb{E}_{X\sim\Xi^*}\left[h(X)\exp\left(s\tilde{h}(X)\right)\right]}{\mathbb{E}_{X\sim\Xi^*}\left[\exp\left(s\tilde{h}(X)\right)\right]}.
\end{aligned}
\tag{98}
$$

Since $s\left\|\tilde{h}\right\|_\infty \le \frac{1}{8}$, we have $s\tilde{h}(X) \in \left[-\frac{1}{4}, \frac{1}{4}\right]$ and therefore the simple inequality

$$\left|\exp(u) - 1 - u\right| \le u^2, \ \forall u \in \left[-\frac{1}{4}, \frac{1}{4}\right],$$

implies

$$
\begin{aligned}
&\mathbb{E}_{X\sim\Xi^*}\left[h(X)\exp\left(s\tilde{h}(X)\right)\right] \\
&\overset{(d)}{\ge} \underbrace{\mathbb{E}_{X\sim\Xi^*}[h(X)]}_{=\,0} + s\mathbb{E}_{X\sim\Xi^*}\left[|h(X)|\left|\tilde{h}(X)\right|\right] - s^2\mathbb{E}_{X\sim\Xi^*}\left[|h(X)|\tilde{h}^2(X)\right] \\
&\overset{(e)}{\ge} s\mathbb{E}_{X\sim\Xi^*}\left[\tilde{h}^2(X)\right] - s^2\sqrt{\mathbb{E}_{X\sim\Xi^*}\left[h^2(X)\right]}\underbrace{\sqrt{\mathbb{E}_{X\sim\Xi^*}\left[h^4(X)\right]}}_{=\,H_{2\to4}\cdot\mathbb{E}_{X\sim\Xi^*}\left[h^2(X)\right]} \\
&\overset{(f)}{\ge} \frac{s}{2}\mathbb{E}_{X\sim\Xi^*}\left[h^2(X)\right] - s^2 H_{2\to4}\left(\mathbb{E}_{X\sim\Xi^*}\left[h^2(X)\right]\right)^{\frac{3}{2}} \\
&= \frac{\|h\|_{L^2(\Xi^*)}}{8}\left(\frac{1}{\sqrt{n}} - \frac{H_{2\to4}}{2n}\right) \\
&\overset{(g)}{\ge} \frac{\|h\|_{L^2(\Xi^*)}}{16\sqrt{n}},
\end{aligned}
\tag{99}
$$

where the step (d) holds due to the fact that $\mathrm{sign}\left(h(x)\right) = \mathrm{sign}\left(\tilde{h}(x)\right), \forall x \in \mathbb{X}$, the step (e) makes use of the property that $\left|\tilde{h}(x)\right| \le |h(x)|, \forall x \in \mathbb{X}$, together with the Cauchy-Schwarz inequality, the step (f) follows due to Lemma 7 in [42], and the step (g) utilizes the assumption that $n \ge 16H_{2\to4}^2$. Putting the lower bound (99) into the equation (98) yields

$$\tau\left(\Xi_s, \Gamma^*\right) - \tau\left(\Xi^*, \Gamma^*\right) \ge \frac{\|h\|_{L^2(\Xi^*)}}{16\sqrt{n}\,\mathbb{E}_{X\sim\Xi^*}\left[\exp\left(s\tilde{h}(X)\right)\right]} \overset{(h)}{\ge} \frac{\|h\|_{L^2(\Xi^*)}}{24\sqrt{n}}, \tag{100}$$

where the step (h) holds since $\mathbb{E}_{X\sim\Xi^*}\left[\exp\left(s\tilde{h}(X)\right)\right] \le \frac{3}{2}$, which follows by the fact $\left|s\tilde{h}(X)\right| \le \frac{1}{8}$. Finally, by taking three pieces (86), (90), and (100) collectively, one completes the proof of Lemma C.1.

## C.3  Proof of Lemma C.2

The proof of Lemma C.2 is also heavily relies on Le Cam's two-point method. For each $(i, s, z) \in [n] \times (0, +\infty) \times \{\pm 1\}$, we consider the function $\mu_i(zs)(\cdot, \cdot) : \mathbb{X} \times \mathbb{A} \to \mathbb{R}$ defined by

$$\mu_i(zs)(x, a) := \mu^*(x, a) + \frac{zsg(x, a)}{\overline{\pi}_i(x, a)} \sigma^2(x, a), \ \forall (x, a) \in \mathbb{X} \times \mathbb{A}. \tag{101}$$

Also, we define the perturbed outcome kernel $\Gamma_i(zs)(\cdot, \cdot) : \mathbb{X} \times \mathbb{A} \to \mathbb{Y}$ as

$$\Gamma_i(zs)(\cdot \,|\, x, a) := \mathcal{N}\left(\mu_i(zs)(x, a), \sigma^2(x, a)\right), \ \forall (x, a) \in \mathbb{X} \times \mathbb{A}.$$

Then, due to Le Cam's two-point lemma, the local minimax risk over the class $\mathcal{C}_\delta(\mathcal{I}^*)$ can be lower bounded by

$$\mathcal{M}_n\left(\mathcal{C}_\delta(\mathcal{I}^*)\right) \geq \frac{1}{4}\left\{1 - \mathrm{TV}\left(\mathbb{P}^n_{(\Xi^*, \Gamma_i(s))}, \mathbb{P}^n_{(\Xi^*, \Gamma_i(-s))}\right)\right\} \\ \left\{\tau\left(\Xi^*, \Gamma_i(s)\right) - \tau\left(\Xi^*, \Gamma_i(-s)\right)\right\}^2, \tag{102}$$

provided that $\Gamma_i(zs) \in \mathcal{N}_\delta(\Gamma^*)$ for $z \in \{\pm 1\}$.

We first upper bound the total variation distance $\mathrm{TV}\left(\mathbb{P}^n_{(\Xi^*, \Gamma_i(s))}, \mathbb{P}^n_{(\Xi^*, \Gamma_i(-s))}\right)$. By employing the Pinsker-Csiszár-Kullback inequality, one has

$$\mathrm{TV}\left(\mathbb{P}^n_{(\Xi^*, \Gamma_i(s))}, \mathbb{P}^n_{(\Xi^*, \Gamma_i(-s))}\right) \leq \sqrt{\frac{1}{2}\mathrm{KL}\left(\mathbb{P}^n_{(\Xi^*, \Gamma_i(s))} \,\middle\|\, \mathbb{P}^n_{(\Xi^*, \Gamma_i(-s))}\right)}. \tag{103}$$

The KL-divergence $\mathrm{KL}\left(\mathbb{P}^n_{(\Xi^*, \Gamma_i(s))} \,\middle\|\, \mathbb{P}^n_{(\Xi^*, \Gamma_i(-s))}\right)$ can be computed as

$$\mathrm{KL}\left(\mathbb{P}^n_{(\Xi^*, \Gamma_i(s))} \,\middle\|\, \mathbb{P}^n_{(\Xi^*, \Gamma_i(-s))}\right)$$
$$= \mathbb{E}_{(\Xi^*, \Gamma_i(s))}\left[\log \frac{p^n_{(\Xi^*, \Gamma_i(s))}(\mathbf{O}_n)}{p^n_{(\Xi^*, \Gamma_i(-s))}(\mathbf{O}_n)}\right]$$
$$= \mathbb{E}_{(\Xi^*, \Gamma_i(s))}\left[\sum_{i=1}^n \log \frac{\xi^*(X_i)\,\pi_i^*(X_i, \mathbf{O}_{i-1}; A_i)\,\gamma_i(s)(Y_i|\,X_i, A_i)}{\xi^*(X_i)\,\pi_i^*(X_i, \mathbf{O}_{i-1}; A_i)\,\gamma_i(-s)(Y_i|\,X_i, A_i)}\right] \tag{104}$$
$$= \sum_{i=1}^n \mathbb{E}_{(\Xi^*, \Gamma_i(s))}\left[\log \frac{\gamma_i(s)(Y_i|\,X_i, A_i)}{\gamma_i(-s)(Y_i|\,X_i, A_i)}\right].$$

Note that

$$\log \frac{\gamma_i(s)(y|\,x, a)}{\gamma_i(-s)(y|\,x, a)}$$
$$= -\frac{1}{2\sigma^2(x, a)}\left[\{y - \mu_i(s)(x, a)\}^2 - \{y - \mu_i(-s)(x, a)\}^2\right] \tag{105}$$
$$= \frac{sg(x, a)}{\overline{\pi}_i(x, a)}\left\{2y - \mu_i(s)(x, a) - \mu_i(-s)(x, a)\right\}.$$

By utilizing the fact (105), one can obtain from the equation (104) that

$$\mathrm{KL}\left(\mathbb{P}^n_{(\Xi^*, \Gamma_i(s))} \,\middle\|\, \mathbb{P}^n_{(\Xi^*, \Gamma_i(-s))}\right)$$
$$= \sum_{i=1}^n \mathbb{E}_{(\Xi^*, \Gamma_i(s))}\left[\mathbb{E}_{(\Xi^*, \Gamma_i(s))}\left[\frac{sg(X_i, A_i)}{\overline{\pi}_i(X_i, A_i)}\left\{2Y_i - \mu_i(s)(X_i, A_i) - \mu_i(-s)(X_i, A_i)\right\}\,\middle|\,(X_i, A_i, \mathcal{H}_{i-1})\right]\right]$$
$$= \sum_{i=1}^n \mathbb{E}_{(\Xi^*, \Gamma_i(s))}\left[\frac{sg(X_i, A_i)}{\overline{\pi}_i(X_i, A_i)}\left\{\mu_i(s)(X_i, A_i) - \mu_i(-s)(X_i, A_i)\right\}\right]$$
$$= 2s^2 \sum_{i=1}^n \mathbb{E}_{(\Xi^*, \Gamma_i(s))}\left[\frac{g^2(X_i, A_i)\,\sigma^2(X_i, A_i)}{\overline{\pi}_i^2(X_i, A_i)}\right] \tag{106}$$

$$= 2s^2 \sum_{i=1}^{n} \mathbb{E}_{\mathcal{I}^*} \left[ \frac{g^2\left(X_i, A_i\right) \sigma^2\left(X_i, A_i\right)}{\overline{\pi}_i^2\left(X_i, A_i\right)} \right]$$

$$\overset{(a)}{\leq} 2K^2 s^2 \sum_{i=1}^{n} \mathbb{E}_{\mathcal{I}^*} \left[ \frac{g^2\left(X_i, A_i\right) \sigma^2\left(X_i, A_i\right)}{\left(\pi_i^*\right)^2\left(X_i, \mathbf{O}_{i-1}; A_i\right)} \right]$$

$$= 2K^2 s^2 n \left\|\sigma\right\|_{(n)}^2,$$

where the step (a) follows by the assumption (25). If we put $s = \frac{1}{2K\sqrt{n}\|\sigma\|_{(n)}}$ into the bound (106), it follows that $\mathrm{KL}\left(\mathbb{P}_{\left(\Xi^*, \Gamma_i(s)\right)}^n \middle\| \mathbb{P}_{\left(\Xi^*, \Gamma_i(-s)\right)}^n\right) \leq \frac{1}{2}$. So, by combining this conclusion together with the basic inequality (103), we arrive at

$$\mathrm{TV}\left(\mathbb{P}_{\left(\Xi^*, \Gamma_i(s)\right)}^n, \mathbb{P}_{\left(\Xi^*, \Gamma_i(-s)\right)}^n\right) \leq \frac{1}{2}. \tag{107}$$

At this point, we should note for every $(i, z, x, a) \in [n] \times \{\pm 1\} \times \mathbb{X} \times \mathbb{A}$ that

$$
\begin{aligned}
\left|\mu^*(x, a) - \mu_i(sz)(x, a)\right| &= \frac{s\left|g(x, a)\right| \sigma^2(x, a)}{\overline{\pi}_i(x, a)} \\
&= \frac{1}{2\sqrt{K}} \cdot \frac{\left|g(x, a)\right| \sigma^2(x, a)}{\sqrt{n}\overline{\pi}_i(x, a) \|\sigma\|_{(n)}} \\
&\overset{(b)}{\leq} \frac{\delta(x, a)}{2\sqrt{K}} \\
&\overset{(c)}{\leq} \delta(x, a),
\end{aligned}
\tag{108}
$$

where the step (b) holds due to Assumption 4, and the step (c) utilizes the fact that $K \geq 1$, which establishes that $\Gamma_i(zs) \in \mathcal{N}_\delta\left(\Gamma^*\right)$ for $z \in \{\pm 1\}$ and thus the local minimax lower bound (102) turns out to be valid.

Next, we aim at establishing a lower bound on the gap between the functional values $\tau\left(\Xi^*, \Gamma_i(s)\right)$ and $\tau\left(\Xi^*, \Gamma_i(-s)\right)$. One can observe that

$$
\begin{aligned}
&\tau\left(\Xi^*, \Gamma_i(s)\right) - \tau\left(\Xi^*, \Gamma_i(-s)\right) \\
&= \mathbb{E}_{X \sim \Xi^*}\left[\left\langle g(X, \cdot), \mu_i(s)(X, \cdot) - \mu_i(-s)(X, \cdot)\right\rangle_{\lambda_\mathbb{A}}\right] \\
&= 2s \cdot \mathbb{E}_{\mathcal{I}^*}\left[\int_\mathbb{A} \frac{g^2\left(X_i, a\right) \sigma^2\left(X_i, a\right)}{\overline{\pi}_i\left(X_i, a\right)} \mathrm{d}\lambda_\mathbb{A}(a)\right] \\
&\overset{(d)}{\geq} \frac{2s}{K} \cdot \mathbb{E}_{\mathcal{I}^*}\left[\int_\mathbb{A} \frac{g^2\left(X_i, a\right) \sigma^2\left(X_i, a\right)}{\pi_i^*\left(X_i, \mathbf{O}_{i-1}; a\right)} \mathrm{d}\lambda_\mathbb{A}(a)\right] \\
&= \frac{2s}{K} \cdot \mathbb{E}_{\mathcal{I}^*}\left[\frac{g^2\left(X_i, A_i\right) \sigma^2\left(X_i, A_i\right)}{\left(\pi_i^*\right)^2\left(X_i, \mathbf{O}_{i-1}; A_i\right)}\right] \\
&= \frac{1}{K^2\sqrt{n}\|\sigma\|_{(n)}} \mathbb{E}_{\mathcal{I}^*}\left[\frac{g^2\left(X_i, A_i\right) \sigma^2\left(X_i, A_i\right)}{\left(\pi_i^*\right)^2\left(X_i, \mathbf{O}_{i-1}; A_i\right)}\right],
\end{aligned}
\tag{109}
$$

where the step (d) holds due to the assumption (25). By taking three pieces (102), (107), and (109) collectively, we have

$$\mathcal{M}_n\left(\mathcal{C}_\delta\left(\mathcal{I}^*\right)\right) \geq \frac{1}{8K^4 n \|\sigma\|_{(n)}^2}\left(\mathbb{E}_{\mathcal{I}^*}\left[\frac{g^2\left(X_i, A_i\right) \sigma^2\left(X_i, A_i\right)}{\left(\pi_i^*\right)^2\left(X_i, \mathbf{O}_{i-1}; A_i\right)}\right]\right)^2 \tag{110}$$

for every $i \in [n]$. Hence, one can conclude by taking an average of the local minimax lower bound (110) over $i \in [n]$ that

$$
\begin{aligned}
\mathcal{M}_n \left( \mathcal{C}_\delta \left( \mathcal{I}^* \right) \right) &= \frac{1}{n} \sum_{i=1}^{n} \mathcal{M}_n \left( \mathcal{C}_\delta \left( \mathcal{I}^* \right) \right) \\
&\geq \frac{1}{8 K^4 n^2 \left\| \sigma \right\|_{(n)}^2} \sum_{i=1}^{n} \left( \mathbb{E}_{\mathcal{I}^*} \left[ \frac{g^2 \left( X_i, A_i \right) \sigma^2 \left( X_i, A_i \right)}{\left( \pi_i^* \right)^2 \left( X_i, \mathbf{O}_{i-1}; A_i \right)} \right] \right)^2 \\
&\overset{(e)}{\geq} \frac{1}{8 K^4 n^3 \left\| \sigma \right\|_{(n)}^2} \underbrace{\left( \sum_{i=1}^{n} \mathbb{E}_{\mathcal{I}^*} \left[ \frac{g^2 \left( X_i, A_i \right) \sigma^2 \left( X_i, A_i \right)}{\left( \pi_i^* \right)^2 \left( X_i, \mathbf{O}_{i-1}; A_i \right)} \right] \right)^2}_{= n^2 \left\| \sigma \right\|_{(n)}^4} \\
&= \frac{1}{8 K^4} \cdot \frac{\left\| \sigma \right\|_{(n)}^2}{n},
\end{aligned}
\tag{111}
$$

where the step (e) makes use of the Cauchy-Schwarz inequality.

