# OpenReview forum: "Off-policy estimation with adaptively collected data: the power of online learning"
_NeurIPS.cc/2024/Conference — NeurIPS 2024 poster_

### Official Review · Reviewer_kxx3 · 2024-07-09

**Soundness:** 3
**Presentation:** 2
**Contribution:** 2
**Rating:** 5
**Confidence:** 4

**Summary:**

This paper presents an approach to estimating linear functionals of the reward function in contextual bandit settings from adaptively collected data. They consider the class of augmented inverse propensity weighted (AIPW) estimators and prove guarantees about the quality of the estimator in terms of the quality of the plugin estimator of the mean reward. Specifically, they characterize the finite sample MSE of the AIPW estimator on adaptively collected data. They quality of the estimator (in terms of low MSE) depends on the quality of the plugin estimator of the mean. They relate online learning of the plugin estimator of the mean reward to online non-parametric regression; they then prove a regret bound, where regret relates to the quality of the plug-in estimator learned online.

***edited score based on rebuttal. See comment below***

**Strengths:**

This paper's presentation is technically precise and seems mathematically thorough. The author's approach of relating the estimation problem online learning seems interesting and creative.

**Weaknesses:**

- There is significant missing discussion of relevant related work on finite sample approaches for off policy evaluation in contextual bandits. For example "Anytime-valid off-policy inference for contextual bandits" by Waubly-Smith et al., "Off-Policy Confidence Sequences" by Karampatziakis et al. and "Optimal and Adaptive Off-policy Evaluation in Contextual Bandits" by Wang et al., as some examples. I would recommend the authors compare to these papers both theoretically and ideally experimentally as well.

- The main result the authors present that holds for general plug-in estimators of the mean is the derivation of the MSE upper bound for the AIPW estimator (Theorem 3.2) as a function of the "regret" of the plug-in estimator error. Beyond this the authors provide results for the plug-in estimators in tabular data settings (Theorem 3.3) and plug-in estimators that are linear models (Theorem 3.4). I feel this is a limited contribution in terms of the types of plug-in estimators the authors can provide guarantees for.

- The main results presented in this work assume that the behavior has a constant exploration rate (Assumption 1). Other works on statistical inference after adaptive sampling generally can prove results when the exploration can decay.

- The writing is technically precise but extremely dense and often lacks sufficient context. I give explicit examples below:

(-) The final result in section 4 was extremely hard to parse. There is a mention of "mis-specification" in the section 4.1 title, but then there is no mention of mis-specification anywhere in the section itself. It is not easy to understand how the result presented in that section is related to mis-specification.

(-) There was little discussion as to why the pertubed IPW estimator was introduced in section 3.1, rather than just starting with the AIPW estimator in 3.2 (since there were no examples discussed of the perturbed IPW estimator that were not some version of an AIPW estimator). Furthermore, if the discussion in 3.1 is kept, the connection between these two estimators presented in 3.1 and 3.2 respectively needs to be made more explicitly. For example, explicitly relating AIPW if for a certain choice of $f$ these two estimators are the same.

**Questions:**

- In line 97, I do not understand why $\mu$ is called a treatment effect. It is also called the "reward model" and appears to be the expected reward function (function of context and action). Why is this called a treatment effect? This is not what is "commonly referred to" as a treatment effect in the causal inference literature.
- In line 107 you state that its assumed the propensities are "revealed", does this mean known?
- Please define << notation used in 117

**Limitations:**

- A severe limitation of this work (especially compared to previous literature) is that they only provide guarantees about the mean squared error of the AIPW estimator and do not provide any approach to constructing confidence intervals for the quantities of interest. I would expect most people interested in using statistical inference methods on adaptively collected data (especially if data is small enough to warrant using a linear plug-in model or be in a tabular data setting) would care solely about estimation and not uncertainty quantification. As a result, the practical utility of this work is very limited.

- There are no simulations demonstrating their approach in practice or comparing to other methods.

---

> ### Author Rebuttal · Authors · 2024-08-05
>
> We thank the reviewer for the detailed reviews. Below are point-by-point responses. We hope the reviewer can increase the score if our responses address your concerns.
>
> 1. Re "There is significant missing discussion of relevant related work on finite sample approaches for off policy evaluation in contextual bandits."
>
> Thank you for the suggested related works. We will discuss these in our paper. Our main goal is different from the suggested papers. The first two papers focus on constructing confidence intervals (or sequences) for the off-policy value, while our objective is in estimation of the off-policy value. The last paper (which we have cited in the paper) focuses on OPE with i.i.d. data, while our work focuses on OPE with adaptively collected data.
>
> Even if we translate the confidence intervals to prediction (in some way), it is hard to compare theoretically as no specific rate is provided for the confidence intervals in the paper “Anytime-valid off-policy inference for contextual bandits” by Waudby-Smith et al. Nevertheless, their methods are interesting, and we will try to compare our method with them empirically.
>
> 2. Re "The main result which holds for general plug-in estimators of the mean reward function."
>
> We note that our main results (Theorems 3.1 and 3.2) hold in general: Theorem 3.1 holds for any sequence of estimates of the treatment effect, and Theorem 3.2 holds for estimates of the treatment effect resulting from any no-regret learning algorithm. Theorems 3.3 and 3.4 are just two examples (i.e., corollaries) of Theorem 3.2 to demonstrate how to apply the framework of online learning for some concrete classes of outcome models. We also provide results for the case of general function approximation in Appendix B.6 that goes beyond the two examples on the tabular setting and linear designs (i.e., the case of linear function approximation).
>
> 3. Re "The main results presented in this paper assume that the behavioral policies have constant exploration rates (Assumption 1)."
>
> Similar to the response above, the strict overlap condition (Assumption 1) is imposed in the examples (Theorems 3.3 and 3.4, and Appendix B.6) of one of our main results---Theorem 3.2. Theorem 3.2 itself is applicable to cases with decaying exploration rates, but for simplicity we didn’t pursue this in this work. We leave this point as future work.
>
> 4. Re "The writing is technically precise, but extremely dense and often lacks sufficient context. "
>
> Thanks for the critique. We will revise the submission accordingly. Here, we give some clarifications on the terminologies “treatment effect” or “mean reward function.” These two terminologies are used separately in two communities: causal inference, and bandits and reinforcement learning. They can be related as seen in line 111 of our paper. This choice of terminology also aligns with a prior work “Off-policy estimation of linear functionals: Non-asymptotic theory for semi-parametric efficiency” by Mou et al.
>
> 5. Re "A severe limitation of this work (especially compared to the existing literature) is that the authors only provide guarantees about the mean-squared error."
>
> We cannot agree with the reviewer on this point. We view the estimation task as equally important as uncertainty quantification (e.g., constructing confidence intervals) from both theoretical and practical perspectives. Moreover, our proposal, the doubly-robust (DR) estimator using an online learning algorithm as a sub-routine, is new to the literature, and practically useful.
>
> 6. Re "There are no simulation results demonstrating their approach in practice or comparing against the other methods in the literature."
>
> We provided preliminary experimental results to corroborate our theory. Please check the attached PDF file. We mostly followed the experimental set-up in the paper “Off-policy evaluation via adaptive weighting with data from contextual bandits” by Zhan et al. Our DR estimator using an online learning algorithm as a sub-routine performs competitively against other estimators, while at the same time enjoying provable finite-sample performance guarantees. We will add more numerical simulation results to understand the pros and cons of each estimator.

---

> > ### Comment · Reviewer_kxx3 · 2024-08-12
> >
> > Hello, I will raise my score a bit. I appreciate the addition of empirical evaluation and the addition of discussion to related work. This addresses bullet 1 of weaknesses I stated. Through rewriting it seems like bullets 2 and 4 can be addressed. My concerns are still bullet 3 and that the amount of edits needed (adding simulations results, and changing writing for bullets 2 and 4) are quite a lot of changes/addition that are not being reviewed.

---

### Official Review · Reviewer_KnWJ · 2024-07-11

**Soundness:** 3
**Presentation:** 2
**Contribution:** 3
**Rating:** 5
**Confidence:** 2

**Summary:**

The paper investigates the challenge of estimating a linear functional of the treatment effect from adaptively collected data, commonly found in contextual bandits and causal inference studies. It introduces finite-sample upper bounds for the mean-squared error (MSE) of augmented inverse propensity weighting (AIPW) estimators and proposes a reduction scheme to minimize these bounds. The method is illustrated through three concrete examples. Additionally, the paper establishes a local minimax lower bound, demonstrating the instance-dependent optimality of the AIPW estimator.

**Strengths:**

-	The paper extends the non-asymptotic theory of AIPW estimators to adaptively collected data.
-	The paper provides both an upper bound and a local minimax lower bound on the MSE of the off-policy value, which quantifies the similarity between a given target evaluation function $g$ and the treatment effect $\mu^*$.

**Weaknesses:**

-	Although the paper is primarily theoretical, it would be helpful if the authors could include some simulation experiments to verify the theoretical results. For example, it would be interesting to see how the regret converges in practice in the examples of Section 3.5.
-	It would be beneficial if the authors could provide more explanations regarding certain definitions. For example, the off-policy value is defined as the expectation of the inner product between $g$ and $\mu^*$, rather than, e.g., the expectation of $g$ itself. What is the rationale behind this definition? In addition, why is the perturbed IPW estimator considered over the traditional IPW estimator?
-	The paper has generalized the theory developed for i.i.d. data to adaptively collected data and discussed the technical difficulties in Section 3.2. Could the authors compare the results for i.i.d. data with those for adaptively collected data? Is there any efficiency loss when the data is collected adaptively?

**Questions:**

See above.

**Limitations:**

Limitations have been discussed in the checklist.

---

> ### Author Rebuttal · Authors · 2024-08-05
>
> We thank the reviewer for the suggestions. Below are the point-by-point responses. We hope the reviewer can increase the score if the weakness listed in the review has been improved.
>
> 1. Re  "it would be helpful if the authors could include some simulation experiments to verify the theoretical results."
>
> We provided preliminary experimental results to corroborate our theory. Please check the attached PDF file. We mostly followed the experimental set-up in the paper “Off-policy evaluation via adaptive weighting with data from contextual bandits” by Zhan et al. Our DR estimator using an online learning algorithm as a sub-routine performs competitively against other estimators, while at the same time enjoying provable finite-sample performance guarantees. We will add more numerical simulation results to understand the pros and cons of each estimator.
>
> 2. Re "It would be beneficial if the authors could provide more detailed explanations regarding certain definitions."
>
> 2.1. Our formulation of the off-policy value using the inner product between the treatment effect $\mu^*$ and the evaluation function $g$ is versatile: As elucidated in lines 109~121 in our paper, our definition of the off-policy value in the equation (1) recovers several important quantities of interest including the average treatment effect (ATE) or its weighted variants and the value function in contextual bandits.
>
> 2.2. The perturbed IPW estimator reduces the variance of the standard IPW estimator. This can be observed from Proposition 3.1 in our paper. With a proper choice of the collection of auxiliary functions $f$, the perturbed IPW estimator has smaller variance.
>
> 3. Re "Could the authors compare the results for i.i.d. data with those for adaptively collected data?"
>
> As seen from both the upper bounds and lower bounds, the main difference lies in the measure of the size of noises: see the equations (7) and (2) in our paper. In an adaptive data collection model, the weighted $\ell_2$-norm of the noise depends on time-varying history-dependent behavioral policies, while in the i.i.d. data collection model, the weight is fixed over time and history-independent: see the paper “Off-policy estimation of linear functionals: Non-asymptotic theory for semi-parametric efficiency” by Mou et al.

---

> > ### Comment · Reviewer_KnWJ · 2024-08-12
> >
> > Thank you for your thoughtful and detailed response to my comments. I have no further questions at this time. As noted by other reviewers, the paper introduces some concepts that are less commonly seen in the literature, making it essential to provide sufficient context and explanation. It appears that the current work is an extension of [1], which indeed provides more background. I recommend including more intuition and context in future revisions to enhance clarity.
> >
> > [1] Mou, W., Wainwright, M.J. and Bartlett, P.L., 2022. Off-policy estimation of linear functionals: Non-asymptotic theory for semi-parametric efficiency. arXiv preprint arXiv:2209.13075.

---

### Official Review · Reviewer_AMTN · 2024-07-19

**Soundness:** 3
**Presentation:** 3
**Contribution:** 3
**Rating:** 5
**Confidence:** 3

**Summary:**

This paper study the off-policy problem in the sequential decision setting with adaptively collected data. The authors propose to use augmented Inverse propensity weighting estimator to estimate the policy value and conduct extensive theoretical analysis on the estimator, including variance and mean square error. Based on the analysis, the authors propose the methods to learn the function estimator in AIPW estimators.

**Strengths:**

This paper comprehensively analyze the property of the AIPW estimator, including the variance and MSE bound. Therefore, I think this is a theoretically solid paper. And the connection between theory and the method is smooth and well-grounded.

**Weaknesses:**

I am confused about the claim of "adaptive" and "online learning" in the title. The two words seems that the decision-maker can adaptively select the action during the decision process. However, it seems that the record (context, actions, outcomes) are passively observed in the problem. So I concern that the paper may be mis-positioned.

Therefore, the technical distinction between the sequential decision setting and static setting is not clearly presented. It seem to be a trivial extention of the traditional policy evaluations problem in the static setting.

**Questions:**

The formulation of  perturbed IPW estimator in section 3.1 seems different with the doubly robust estimator.  Can the authors provide the connection between the perturbed IPW estimator and the doubly robust estimator in other papers. For example, the doubly robust estimators in [1].

[1] Miroslav Dudik, John Langford, and Lihong Li. 2011. Doubly robust policy evaluation and learning. In International Conference on International Conference on Machine Learning. 1097–1104.

**Limitations:**

I think the discussion of the limitation in the paper is not sufficient.

---

> ### Author Rebuttal · Authors · 2024-08-05
>
> We thank the reviewer for the comments. Below are our point-by point responses to better position our paper and explain our contributions. We hope the reviewer can increase the score if the confusion about the paper and its contributions has been resolved.
>
> 1. Re "I am confused about the claim of "adaptive" and "online learning" in the title."
>
> a) Why "adaptive" in the title? We focus on off-policy evaluation (OPE): the problem of estimating the value function based on an offline dataset. In OPE, the word “adaptive” means that in the offline dataset, the actions could be adaptively chosen based on previous samples. This is in contrast to OPE with i.i.d. dataset in which actions are drawn depending only on the current context.
>
> b) Why "online learning" in the title? We use an “online learning” algorithm (or a no-regret learning algorithm) to iteratively learn the treatment effect function, and then use them in forming the AIPW estimator to solve the OPE problem; see Algorithm 1 in the paper. This is also different from OPE with i.i.d. dataset in which an “offline learning” algorithm such as the empirical risk minimization (ERM) is applied to learn the treatment effect. This gives rise to the term “the power of online learning” in the title.
>
> 2. Re "the technical distinction between the sequential decision setting and static setting is not clearly presented."
>
> We hope the response above has already resolved some confusion about sequential vs static. We expand a bit on the challenge and our contributions here.
>
> We view OPE as a static decision-making problem as the pre-collected offline dataset is already provided to the learner, and the learner's goal is to estimate the off-policy value. The major challenge here is that the offline data is adaptively collected, that is, in the sequence of observations of the form (context, action, outcome), the actions can be chosen depending on previous data. Focusing on OPE with adaptively collected data, we investigate finite-sample guarantees of the class of AIPW estimators, propose a specific AIPW estimator with a sequence of estimates of the mean reward function learned via an online learning algorithm, and show its near-optimality in a minimax sense. All of these are new to the literature.
>
> 3. Re "The formulation of the perturbed IPW estimator in Section 3.1 seems to be different from the doubly-robust (DR) estimator. "
>
> The AIPW estimator (i.e., the DR estimator) is a special case of the perturbed IPW estimator with the collection of auxiliary functions $f^*$ takes form as the equation (5) in our paper, and the treatment effect $\mu^*$ is replaced by its empirical estimates. This will yield the DR estimator (cf. Eq (1)) in the paper “Doubly robust policy evaluation and learning” by Dudik et al.

---

> > ### Comment · Reviewer_AMTN · 2024-08-13
> >
> > Thanks for your response. I have get the meaning of "adaptive" and "online learning". However, I am still confused that why this problem needs the technology of online learning. Why ERM fails in this setting? Can you give me an example in practice to verify the practical significance？

---

> > > ### Author Response · Authors · 2024-08-13
> > >
> > > We would like to thank the reviewer for engaging in the discussion. The classical empirical risk minimization (ERM) is not an appropriate strategy for estimation of the treatment effect $\mu^*$ in our setting for the following two reasons.
> > >
> > > First, we would argue that online learning is a more natural strategy for estimation of the treatment effect $\mu^*$ in our setting, compared to the classical ERM. In view of Theorem 3.1 in our paper, we need to aim at building a sequence of estimates of the treatment effect $\mu^*$ that minimizes the weighted average estimation error in the equation (11). This objective naturally falls into the realm of online learning, where a sequence of decisions are made to minimize a sequence of certain loss functions. On the contrary, for the i.i.d. data collection model, one only needs to construct a single estimate $\hat{\mu}$ of the treatment effect $\mu^*$ that minimizes a certain weighted mean-squared error; see the equation (11) in [1] for the construction of the estimate. In this case, the classical ERM (the non-parametric weighted least-squares estimate for this case) is more natural.
> > >
> > > Second, one could consider using the ERM in each step of the framework of online learning, i.e., an “adaptive” ERM. However, it is known that this algorithm may incur a linear regret in the worst case, which motivates us to employ no-regret learning algorithms such as the Follow-The-Regularized-Leader (FTRL; basically using the regularized ERM in each step) or its optimistic variants.
> > >
> > > [1] Wenlong Mou, Martin J. Wainwright, and Peter L. Bartlett, “Off-policy estimation of linear functionals: Non-asymptotic theory for semi-parametric efficiency”, arXiv preprint arXiv:2209.13075, 2022.

---

### Author Rebuttal · Authors · 2024-08-07

Please check the attached PDF file for preliminary experimental results to corroborate our theory in the paper.

---

### Decision · Program_Chairs · 2024-09-25

**Decision:**

Accept (poster)

**Comment:**

This is a significant piece of theory work that presents finite sample bound for AIPW estimators. While concerns have been raised regarding related work, clarity of writing and lack of experimental evaluations, all of them have been addressed during the rebuttal period. Therefore I would recommend acceptance.